# Variant antigen diversity in *Trypanosoma vivax* is not driven by recombination

Sara Silva Pereira [1], Kayo J.G. de Almeida Castilho Neto[2], Craig W. Duffy [1], Peter Richards[1], Harry Noyes[3], Moses Ogugo[4], Marcos Rogério André[2], Zakaria Bengaly[5], Steve Kemp[4], Marta M.G. Teixeira[6], Rosangela Z. Machado[2] & Andrew P. Jackson[1✉]

African trypanosomes (*Trypanosoma*) are vector-borne haemoparasites that survive in the vertebrate bloodstream through antigenic variation of their Variant Surface Glycoprotein (VSG). Recombination, or rather segmented gene conversion, is fundamental in *Trypanosoma brucei* for both *VSG* gene switching and for generating antigenic diversity during infections. *Trypanosoma vivax* is a related, livestock pathogen whose *VSG* lack structures that facilitate gene conversion in *T. brucei* and mechanisms underlying its antigenic diversity are poorly understood. Here we show that species-wide *VSG* repertoire is broadly conserved across diverse *T. vivax* clinical strains and has limited antigenic repertoire. We use variant antigen profiling, coalescent approaches and experimental infections to show that recombination plays little role in diversifying *T. vivax VSG* sequences. These results have immediate consequences for both the current mechanistic model of antigenic variation in African trypanosomes and species differences in virulence and transmission, requiring reconsideration of the wider epidemiology of animal African trypanosomiasis.

[1] Department of Infection Biology, Institute of Infection and Global Health, University of Liverpool, 146 Brownlow Hill, Liverpool L3 5RF, UK. [2] Department of Veterinary Pathology, Faculty of Agrarian and Veterinary Sciences, São Paulo State University (UNESP), Jaboticabal, SP, Brazil. [3] Institute of Integrative Biology, University of Liverpool, Biosciences Building, Crown Street, Liverpool L69 7ZB, UK. [4] Livestock Genetic Programme, International Livestock Research Institute, 30709 Naivasha Road, Nairobi, Kenya. [5] International Research Centre for Livestock Development in the Sub-humid Zone (CIRDES), No. 559, rue 5-31 angle, Avenue du Gouverneur Louveau, Bobo-Dioulasso, Burkina Faso. [6] Department of Parasitology, Institute of Biomedical Sciences, University of Sao Paulo, Avenue Professor Lineu Prestes, 1374 Cidade Universitaria, Sao Paulo, SP 05508-000, Brazil. ✉email: a.p.jackson@liv.ac.uk

African trypanosomes (*Trypanosoma* spp.) are unicellular haemoparasites and the cause of African Trypanosomiasis in animals and humans[1]. These parasites are transmitted by tsetse flies (*Glossina* spp.), and their proliferation in blood and other tissues leads to anaemia, immune and neurological dysfunction, which is typically fatal if untreated. The profound, negative impact of this disease on livestock productivity across sub-Saharan Africa is measured in billions of dollars annually[2].

*Trypanosoma vivax* is a livestock parasite found throughout sub-Saharan Africa and South America[3–5]. Although superficially like the more familiar *Trypanosoma brucei* (the species responsible for Human African trypanosomiasis) and *Trypanosoma congolense* (another livestock parasite), *T. vivax* is distinct in morphology and motility[6], cellular ultrastructure[7,8] and genetic repertoire, particularly with regard to cell surface-expressed genes[9,10]. Most conspicuously, it has a simpler life cycle in tsetse flies, lacking a procyclic stage in the insect midgut, and can be transmitted non-cyclically by other genera of haematophagous flies[6].

Although distinct from *T. brucei*, *T. vivax* shares a defining phenotype with other African trypanosomes. Trypanosome cell surfaces are coated with a variant surface glycoprotein (VSG) that undergoes antigenic variation[11]. Trypanosome genomes encode hundreds of alternative *VSG*, but each cell expresses just a single variant. Periodically, new variants emerge that have dynamically switched to an alternative expressed *VSG*[11]. Each VSG is strongly immunogenic but confers no heterologous protection. Thus, as antibodies clear the dominant VSG clones of the parasite infra-population, serologically distinct clones replace them, rendering cognate antibodies redundant and facilitating a persistent infection[12].

Previously, we showed that *T. vivax VSG* are distinct from those in *T. brucei* or *T. congolense*. *T. vivax VSG* genes (named Fam23–26 inclusive) display much greater sequence divergence, and include sub-families absent in other species[13]. In *T. brucei*, recombination in the form of segmental gene conversion (SGC) is instrumental in both switching *VSG* genes and generating novel mosaic antigens[14,15]. However, sequence repeats known to facilitate gene conversion in *T. brucei* were absent from the *T. vivax* reference genome, suggesting that the *T. brucei*-based paradigm of antigenic variation might not apply to other species[10].

Experiments from the pre-genomic era revealed certain distinct features of antigenic variation in *T. vivax* that remain unexplained. Animals infected with *T. vivax* self-cure more often and faster compared with other species, which was attributed to antigenic exhaustion[16,17]. Clones expressing certain VSG re-emerged late in infection after the host had developed immunity[3,17]. Quite unlike *T. brucei* or *T. congolense*, recovered animals displayed immunity to strains from very distant locations, indicating that *T. vivax* serodemes could span countries, or even the whole continent[18,19]. Such features prompted the prediction that antigen repertoire in *T. vivax* would be smaller than in other trypanosomes[3].

Here, we address these long-standing issues by characterising antigenic diversity in clinical *T. vivax* isolates. We apply the data to examine *VSG* recombination in parasite populations and to profile *VSG* expression during experimental infections in a goat model. The variant antigen profile (VAP) we establish for *T. vivax* shows that *VSG* sequence patterns in *T. vivax* are incompatible with the current, *T. brucei*-based model for antigenic variation in trypanosomes.

## Results

**Genome sequencing.** Genomes of 28 *T. vivax* clinical strains isolated from seven countries were sequenced on the Illumina MiSeq platform. Genome assemblies ranged in coverage from 32.8 to 80.4%, in-sequence depth from 3.5× to 78.5×, and in contiguity (N50) from 238 to 2852 (Supplementary Data 1). Using sequence homology with known *VSG* sequences in the *T. vivax* Y486 and *T. brucei* TREU927 reference genomes, between 40 and 436 *VSG* genes were recovered from assembled genome contigs; the mean average (175) is approximately one fifth of the *T. vivax* Y486 reference genome repertoire ($N = 865$)[10].

**T. vivax variant antigen profiles reflect genealogy.** We devised a VAP for *T. vivax VSG* gene repertoire to examine antigenic diversity across strains. The four *VSG*-like gene sub-families (Fam23–26)[13] in the *T. vivax* Y486 reference sequence (hereafter called 'Y486') occurred in all genomes, in similar proportions (Supplementary Fig. 1), making them unsuitable for discriminating between strains. Therefore, we produced clusters of orthologous genes (COGs) for all *VSG*-like sequences from Y486 and 28 clinical strains ($N = 6235$), defining a COG as a group of *VSG*-like sequences with ≥90% sequence identity. This produced 2039 COGs, each comprising a single gene plus near-identical paralogues from multiple strains. Most COGs (78%) were cosmopolitan (i.e. present in multiple locations; see Methods), while 441 were strain-specific (Supplementary Data 2).

VAPs based on presence or absence of *VSG* COGs were compared to strain genealogy and geography to examine spatio-temporal variation in *VSG* repertoire. Figure 1 shows that strain genealogy estimated from whole-genome single nucleotide polymorphisms (SNPs) recapitulates geography and matches the relationships inferred from the VAPs at a regional level, although there are inconsistencies in strain relationships, for instance in the position of 'TvGondo' and 'TvMagna', which may reflect sampling error within the SNP tree or ancestral gene flow between *T. vivax* populations. Overall, VAP broadly reflects both population history and location. There is a remarkable correspondence between VAPs of Ugandan strains with those from Brazil, suggesting that these Brazilian *T. vivax* were introduced into Brazil from East Africa. The correspondence of VAPs and SNPs is particularly clear when we compare the Ugandan/Brazilian profile with those in Nigeria. Still, while clearly divergent in their *VSG* repertoire, there remain 769 COGs (37%) that are shared between these distant locations; for instance, 'TvILV-21' possesses various COGs widespread in West Africa. Thus, *T. vivax VSG* repertoires diverge in concert with the wider genome and provide a faithful record of population history, in contrast to *T. congolense*, where the opposite effect was observed[20].

**Species-wide T. vivax VSG repertoire contains 174 phylotypes.** The *VSG* gene complements in our strain genome sequences are incomplete. So, while comparing partial strain genomes in combination provides a coherent analysis of species-wide *VSG* variation, the spatial distribution of COGs, and the number of truly location-specific COGs, will increase with greater sampling. This is clear when we consider that 248 COGs (12.2%) comprise a single Y486-specific sequence, which is the only strain with a complete *VSG* complement. Presently, a COG-based VAP will include too many false-negative 'absences' to reliably profile individual strains.

A VAP that allows comparison of any two strains must be based on universal markers that also vary in the population. COGs are not universal and sub-families do not vary; so, we reasoned that a taxon of intermediate inclusivity would satisfy both criteria. Therefore, we devised another VAP based on phylotypes, each consisting of multiple, related COGs with ≥70% sequence identity (see Methods and Supplementary Fig. 2 for further explanation of classification system). In all, 174 *VSG*

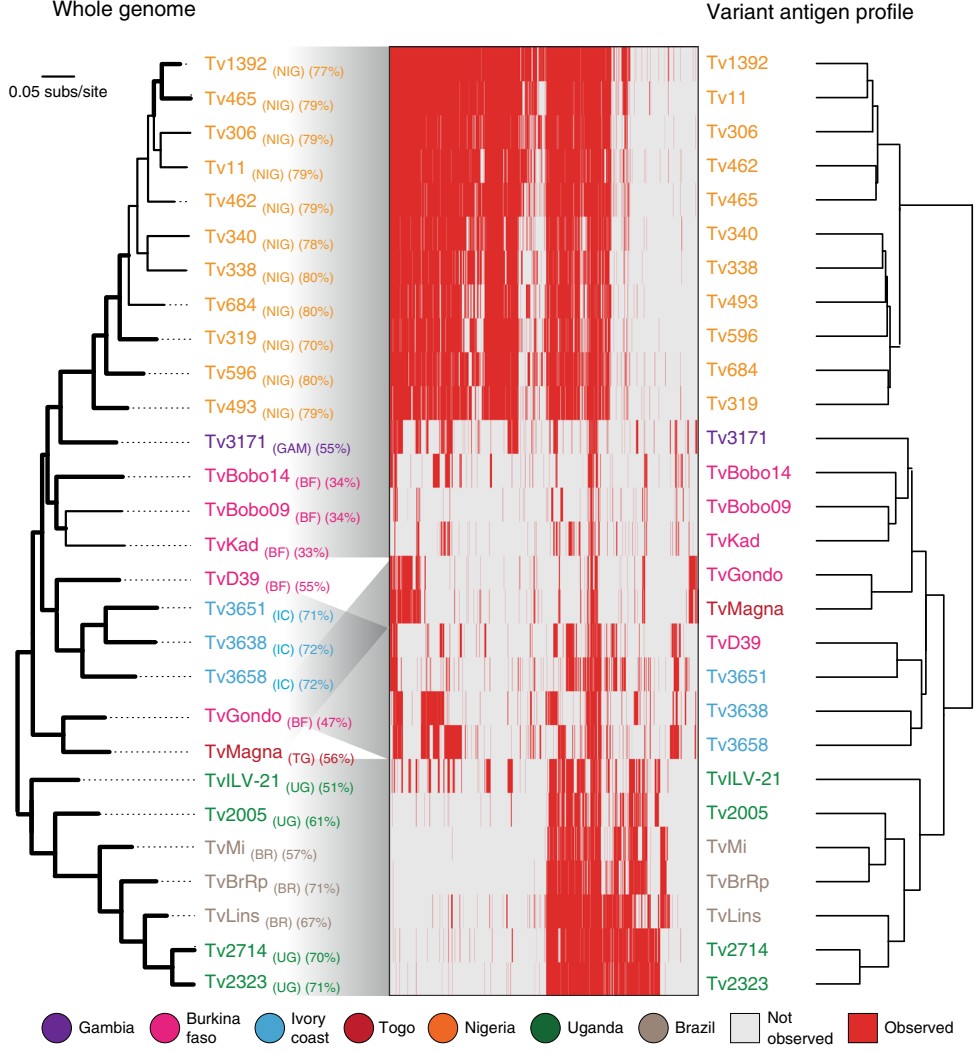

**Fig. 1 Variant antigen profiles of *T. vivax* clinical isolates based on presence and absence of *VSG* gene clusters are concordant with population history (i.e. genetic relatedness).** On the left, a Maximum Likelihood phylogenetic tree estimated from a panel of 21,906 whole-genome SNPs using a GTR + Γ + I model. Branch support is provided by 100 bootstrap replicates and branches with bootstrap support >70 are shown in bold. Percentage genome coverage is shown for each strain in brackets following its label. Genome sequence reads for 28 *T. vivax* clinical strains were mapped to 2038 *VSG* type sequences, representing conserved clusters of orthologous genes (COGs) or strain-specific sequences, to determine the distribution of each *VSG*. Presence (red) or absence (white) of each *VSG* in each strain is indicated in the central panel. Each profile is labelled with the strain name, coloured by its geographical origin, and linked to the SNP tree by the grey shade. On the right, a dendrogram relating all strains according to their observed *VSG* repertoire was estimated from Euclidean distances between VAPs. Source data are provided as a Source Data file.

phylotypes accommodated every *VSG*-like sequence we observed. Figure 2 shows the size and distribution of these across strains and emphasises the widespread distribution of most phylotypes, 86% (149/174) of which are cosmopolitan.

Exceptions to this trend, i.e. structurally distinct *VSG* sub-families restricted to specific populations, may be epidemiologically important. Among Nigerian samples, the location with the largest sample (N = 11) and so the most robust presence/absence calls, five phylotypes are unique (P94, P118, P126, P170, P173). These are not recent derivations in Nigerian *T. vivax* because they are defined by a threshold sequence identity and so are of approximately equally age to other phylotypes. Moreover, their positions in Fig. 2 indicate no significant difference in the node connectivity of Nigeria-specific and cosmopolitan phylotypes overall. As these phylotypes comprised only one or two COGs, we extended the analysis to COGs generally.

We found 130 COGS in at least 9/11 Nigeria strains and no other location. We tested whether the Nigeria-specific COGs were

as old as closely related cosmopolitan COGs in the same phylotype or otherwise evolved more recently. If they were younger than the cosmopolitan COGs, this would mean that, at least in Nigeria, novel VSGs were being generated through gene duplication. Thus, we estimated Maximum Likelihood phylogenies for each phylotype containing a Nigerian-specific COG and inferred relative divergence times using the RelTime tool in MEGA v10.0.5[21] (see Source Data file). This showed that there was no significant difference (p = 0.35, independent *t* test) in the mean divergence times for Nigeria-specific COGs ($\mu = 0.038 \pm 0.005$; N = 83) and cosmopolitan COGs in the same phylotype ($\mu = 0.041 \pm 0.005$; N = 212). Therefore, Nigeria-specific COGs and phylotypes are just as ancient as lineages with cosmopolitan distributions, and do not provide evidence for population-specific gene family expansions.

In summary, the incompleteness of strain genomes compelled us to adopt phylotypes as a universal but variable metric to profile *T. vivax VSG* repertoire. On this basis, *T. vivax VSG* repertoire

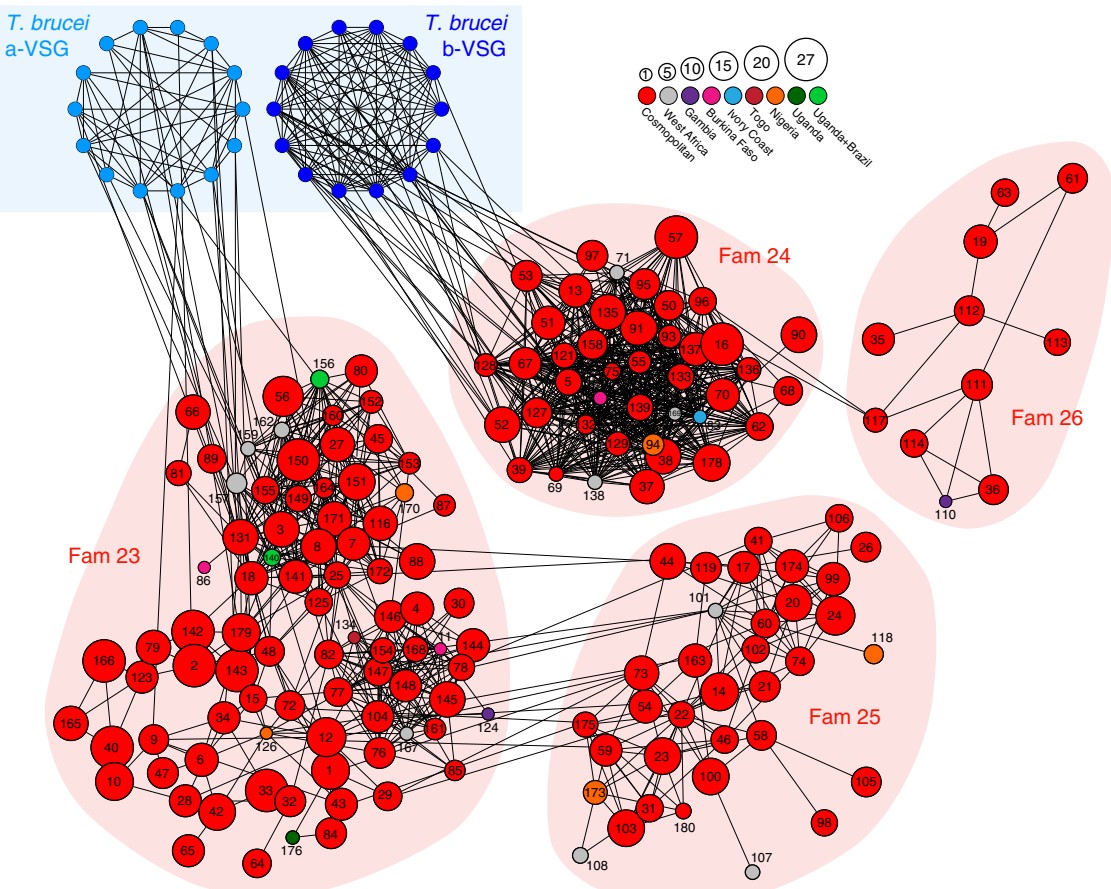

**Fig. 2 The *T. vivax* VSG repertoire is described by 174 phylotypes.** A sequence homology network in which nodes represent phylotypes. Four conserved *VSG* sub-families (Fam23–26[13]) are indicated by pale red back-shading. Nodes are labelled by phylotype number; node size indicates the number of COGs in each phylotype, while node colour indicates the geographical distribution of the phylotype across 28 clinical isolates. Edges represent PSI-BLAST similarity scores greater than a threshold necessary to connect all phylotypes within sub-families. Structural homology of Fam23 and Fam24 with A-type and B-type *T. brucei VSG* respectively is indicated at top left. The figure shows that most phylotypes are cosmopolitan in nature, found in multiple strains and in more than two regions. A minority are strain- or location-specific phylotypes, e.g. there are ten phylotypes specific to West Africa (i.e. Ivory Coast, Togo and Burkina Faso) and another 15 phylotypes that are unique to a single location, for instance five in Nigeria (P94, P118, P126, P170, P173), three in Burkina Faso (P11, P86, P120) and two in The Gambia (P110, P124). Source data are provided as a Source Data file.

appears to be relatively conserved over large distances. Population variation does exist, especially at COG level, but appears to originate through differential patterns of lineage loss rather than population-specific gene family expansions, since Nigeria-specific COGs are no younger than other *VSG*. This degree of widespread conservation is quite unlike patterns seen in *T. brucei*[22]. Suspecting that this indicated a more fundamental difference between African trypanosome species in how antigenic diversity evolves, we examined population variation among their *VSG* sequences in detail.

**Minimal signature of recombination in *T. vivax* VSG sequences.** We took multiple approaches to test the hypothesis[10] that *T. vivax VSG* recombine less than *T. brucei* and *T. congolense VSG*. First, we asked if *VSG* sequences assort. Based on the current model of antigenic switching[11], *VSG* reads from 28 clinical strains would not remain paired after mapping to Y486 because historical recombination events would have distributed them across multiple reference loci. Figure 3a shows that the proportion of strain read-pairs remaining paired after mapping is significantly higher in *T. vivax* (mean = 92%; $N = 19$) relative to *T. congolense* (mean = 87%; $t = 3.23$; $p < 0.05$, independent *t* test) and *T. brucei* (mean = 76%; $t = 12.8$; $p < 0.001$, independent *t* test), and is

almost as high as a negative control comprising adenylate cyclase genes (mean = 97%).

Reversing this approach, we examined how Y486 *VSG* gene sequences mapped to strain assemblies when broken into 150 bp segments. From this segmental mapping, we characterised *VSGs* into fully coupled (FC), multi-coupled (MC), or uncoupled (UC), based on how many reference donors each *VSG* had and how much of the *VSG* sequence they accounted for (Fig. 3b). FC *VSGs* have at least one donor contributing to >84% of the sequence; MC *VSGs* are sequences with donor(s) contributing to less than 84% of the sequence but more than 150 bp, or at least two donor fragments in different regions; and UC *VSGs* are those with one or more donors contributing with one fragment only (i.e. ≤150 bp). The mean proportion of Y486 *VSG* that are mosaics of strain genes (i.e. 'Multi-coupled' (MC: 25%) or 'Uncoupled' (UC: 7%)) is significantly lower than for equivalent comparisons in *T. congolense* (MC: 33%, $p < 0.05$; UC: 31%, $p < 0.001$; independent *t* test) and *T. brucei* (MC: 39%, $p < 0.001$; UC: 12%, $p < 0.001$; independent *t* test), while the number that are essentially orthologous (i.e. 'Fully coupled' (FC: 59%)) is significantly greater (for *T. congolense*, $p < 0.001$; for *T. brucei*, $p < 0.001$; independent *t* test) (Fig. 3c). Analysis of phylogenetic incompatibility in alignments of FC and MC quartets using PHI[23] corroborates the

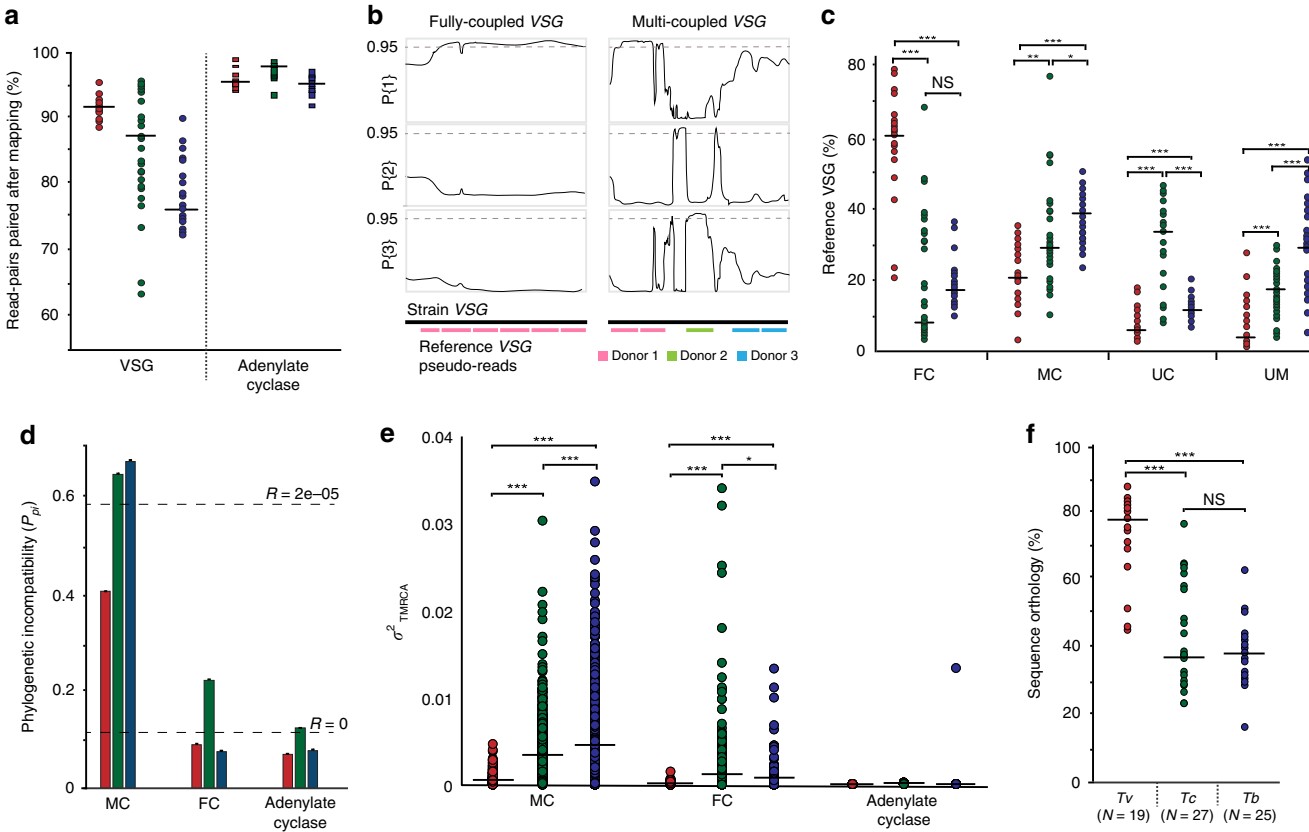

**Fig. 3 The frequency of *VSG* recombination differs between African trypanosome species. a** The proportion of read-pairs from strain *VSG* remaining paired after being mapped to the reference sequence for each trypanosome genome, shaded by species. Adenylate cyclase genes (AC) were included as a negative control. **b** The definition of fully coupled (FC) and multi-coupled (MC) *VSG* sequences. Reference *VSG* sequences were segmented and mapped to strain VSGs. Where ≥85% of pseudo-reads map to the same locus (e.g. 'Donor 1'), the gene is fully coupled. Where a strain VSG has multiple segments mapping to multiple locations (e.g. 'Donor 1–3'), the gene is multi-coupled. Example *T. brucei VSG* sequence quartets are shown after TOPALi HMM analysis[82] (see Methods). The three line graphs represent the Bayesian probabilities of three possible topologies for a quartet phylogeny. An FC *VSG* displays the same topology along its whole length. An MC *VSG* displays different phylogenetic signals along its length, dependent on the identity of the sequence donor. **c** A comparison of the proportions of FC, MC, uncoupled (UC) and unmapped (UM) *VSG* in each trypanosome species. The median value is shown as a black bar. Statistical significance of differences in the mean are indicated by asterisks (independent *t* test, *$p < 0.05$; **$p < 0.01$; ***$p < 0.001$). **d** Phylogenetic incompatibility among *VSG* genes using Phi[23]. The proportion of FC and MC *VSG* quartet alignments showing significant phylogenetic incompatibility ($P_{pi}$) in MC and FC *VSGs* is shown, shaded by species (mean ± s.e.m.). Observed $P_{pi}$ values for simulated sequences generated by NetRecodon[78], either with recombination ($R = 2e^{-05}$) or without ($R = 0$), are indicated by dashed lines. **e** Variation in the 'time to most recent common ancestor' (TMCRA) along MC and FC *VSG* quartet alignments, estimated from ancestral recombination graphs constructed by ACG[81]. The median value is shown as a black bar. **f** Total sequence orthology among *VSG* repertoires in each species. Orthology was calculated as the proportion of *VSG* base-pairs fully coupled between each strain genome sequence and the reference. Number of strain genomes is shown in brackets. Source data are provided as a Source Data file.

mapping patterns. Across all species, FC *VSG* contain little evidence for phylogenetic incompatibility and not generally more than the adenylate cyclase control (Fig. 3d). While MC *VSG* display phylogenetic incompatibility, *T. vivax* MC quartets displayed this less frequently ($P_{pi} = 41\%$) than in *T. congolense* ($P_{pi} = 65\%$) and *T. brucei* ($P_{pi} = 67\%$).

While there are fewer MC *VSG* in *T. vivax*, this sizeable minority might still be genuine mosaics. Alternatively, other processes such as gene paralogy or substitution rate heterogeneity could account for the signature of recombination. Hence, we explicitly modelled the history of recombination within FC or MC sequence quartets using ancestral recombination graphs (ARG) and inferred the time to most recent common ancestor (TMRCA) for each quartet. Average TMRCA was significantly greater for *T. vivax* FC *VSG* (0.19 ± 0.17) than either *T. congolense* (0.05 ± 0.06) or *T. brucei* (0.06 ± 0.07), indicating much deeper coalescent times for *T. vivax VSG*. More importantly, the variance in TMRCA along sequence alignments is extremely small for *T. vivax* FC *VSG*, showing that the

whole alignment shares a common ARG (Fig. 3e). Variance is greater for MC *VSG*, but both MC and FC types are significantly less variable than either other species ($p < 0.001$, independent *t* test). Both the relatively small TMRCA and variance in TMRCA along alignments indicates that *T. brucei* and *T. congolense VSG* are routinely mosaics, while the coalescence of most *T. vivax VSG* can be modelled without recombination. Interestingly, TMRCA variance is significantly higher among *T. brucei* MC *VSG* quartets than *T. congolense VSG* ($p < 0.001$, independent *t* test), indicating that the former may have a higher recombination rate (explored further in Supplementary Table 1).

In summary, these analyses show that retention of orthology among *VSG* loci across trypanosome populations varies significantly between species. Figure 3f plots the total pairwise orthology between strains (see Methods). Around 75% of *T. vivax VSG* are found in multiple strains as orthologues, without evidence for recombination, compared with ~40% in *T. brucei* ($p < 0.001$, independent *t* test) and *T. congolense* ($p < 0.001$, independent

*t* test). As the VAPs indicated, *T. vivax VSG* typically retain orthology and essentially behave like 'normal' genes in the population, while *T. brucei* or *T. congolense VSG* recombine frequently, causing loss of orthology and the appearance of strain-specific mosaics throughout the population.

**Strong phylogenetic effects on *VSG* expression in vivo.** Broadly conserved VSG phylotypes containing little signature of historical recombination indicate that *VSG* mosaics do not contribute to antigenic diversity in vivo. We tested this by measuring *VSG* transcript abundance in goats experimentally infected with *T. vivax* (strain Lins[24]) over a 40-day period. Parasitaemia and expression profiles of VSG phylotypes in four replicates are shown in Fig. 4. We observed the expected waves of parasitaemia beginning after 4 days and continuing approximately every 3 days until termination (i.e. 6–9 parasitaemic peaks). Transcriptomes were prepared for each peak and revealed 282 different *VSG* transcripts across all replicates (Supplementary Data 3), which belonged to 31 different phylotypes (18% of total species repertoire).

Variant antigen profiling of the expressed transcripts characterised the dominant (but more often co-dominant) VSG phylotypes across successive peaks (Fig. 4). Somewhat contrary to expectation, we often saw persistent expression of a phylotype (P) across peaks, e.g. P24 (Supplementary Fig. 3) and P2 (Supplementary Fig. 4), or re-emergence of a phylotype after decline, e.g. P40 (Supplementary Fig. 5) and P143 (Supplementary Fig. 6). The *T. vivax* (Lins) inoculum was not derived from a clone, but rather represents a mixed population with one dominant clone (see Methods); hence, this initial heterogeneity could result in variation in VSG expression between animals. However, despite the unavoidable clonal mixture of the initial inoculum, the identity of expressed phylotypes was partly reproduced across replicates, with 12/31 phylotypes observed in all four animals, and 19 phylotypes in three animals (Supplementary Fig. 7); on 21 occasions this extended to an identical *VSG* sequence (for detail, see Supplementary Figs. 3–6).

Similarly, the order of *VSG* expression was partly reproducible across animals. Figure 5 displays transcript number and abundance at early, middle and late points in the experiment, mapped on to the sequence similarity network of all phylotypes. The best example of reproducibility is the dominant expression of P24 in the middle-to-late period across all animals. Other examples include a group of phylotypes (P2, P40, P142 and P143) expressed early (i.e. peak 1/2, Fig. 5a) in Animal (A) 2 and A3, then re-emerging later at peak 5/6 in A1–3 (Fig. 5b), and even later in A4. For detailed analysis of phylotype abundance at each time-point, see Supplementary Fig. 8. Importantly, however, while phylotypes show consistency in expression through time and across replicates, individual *VSG* transcripts do not. Hence, while the dominant variant antigen belonged to P24 in every replicate, the actual P24 transcript expressed was different in each case and its variants diverged by up to 26.5% in nucleotide identity. Further examples in Supplementary Figs. 3–6 demonstrate that this was typical.

Across all peaks, groups of related transcripts of the same phylotype were commonly co-expressed at the same peak (e.g. P2 expression comprised $3.08 \pm 1.97$ transcripts on average, P24 = $2.33 \pm 1.3$, P40 = $2.67 \pm 1.12$, P143 = $2.71 \pm 1.25$). On three occasions, the observed phylotype comprised seven distinct transcripts (P2 at peak 5 in A1, P8 at peak 8 in A4 and P135 at peak 5 in A1). Overall, only 8/31 phylotypes were ever represented by a single transcript. This indicates that the expressed repertoire is determined in part by sequence homology, and Supplementary Fig. 9 shows that expressed transcripts belong to significantly

fewer phylotypes than simulated transcript repertoires of the same size, confirming that they are not drawn from the available repertoire by chance. For detailed examples, see Supplementary Figs. 3–6.

An obvious feature in Fig. 5 is the concentration of highly expressed phylotypes in the bottom-left corner of the network. A complex of closely related Fam23 phylotypes (e.g. P2, P40, P142) were expressed early in A1 and A2 (Fig. 5a, b). This was followed by Fam23 phylotypes placed closer to the centre of the network (e.g. P8), and finally, Fam25 phylotypes (e.g. P24/P44) in late infection. In A3 and A4, a similar pattern occurred, except that Fam25 *VSG* (i.e. P44) were expressed early, followed by the Fam23 'complex' and then P24. This can also be seen in Supplementary Fig. 8, where phylotypes displaying reproducible profiles across replicates are often closely related (e.g. P2, P40, P142 and P143). The connectivity of nodes representing expressed phylotypes is greater than that expected by chance. The clustering coefficient of a sub-network representing all 'expressed' nodes across all peaks is significantly greater than randomised sub-networks of the same size (for detail, see Supplementary Fig. 10).

In summary, the major pattern emerging from in vivo expression profiles is a strong phylogenetic signal on three levels. First, the identity and order of expressed phylotypes is partly reproducible (but expression of individual transcripts is typically not). Second, phylotypes expressed at a given peak regularly comprise multiple distinct, but closely related, transcripts. Finally, at the phylotype level, related phylotypes are expressed simultaneously or consecutively, manifested as clustering in Fig. 5 and Supplementary Fig. 8. Therefore, phylogeny (or sequence relatedness) is an important factor in explaining the pattern of *VSG* expression during these *T. vivax* infections.

**No mosaics of VSG phylotypes during experimental infections.** Expressed *VSG* in *T. brucei* include sequence mosaics, which is interpreted as evidence for recombination of *VSG* loci during infections[15,25,26]. In *T. brucei*, *VSG* mosaics can be formed between highly divergent donors with as little as 25% identity along their entire lengths[26], and can implicate relatively short recombinant tracts of ~100 bp[27]. We analysed expressed *VSG* transcript sequence mosaics by comparing 100 bp windows of each transcript to the *T. vivax* Lins genome sequence using BLASTp[28]. Typically, mosaics would be confirmed where a single transcript displayed affinities to different *VSG* genes along its length. Unfortunately, since both *VSG* transcripts and gene sequences were often fragmentary, it was common for a transcript to have multiple affinities as no single gene sequence spanned its length. Even so, without exception, the closest related sequences in every window of each transcript were other sequences in the same phylotype.

With sequence affinities inconclusive, we searched for reorganisation of an expressed *VSG* sequence relative to a genomic locus by mapping all read-pairs belonging to *VSG* transcripts to the *T. vivax* Lins genome. After mapping, the percentage of unpaired reads (1.06–5.63%) was greater than the percentage arising from a random selection of 100 housekeeping genes (0.01–0.05%). However, given that *T. vivax VSG* are arranged in tandem gene arrays of closely related paralogues[10], we reasoned that this repetitive organisation might lead to multiple mapping of reads. Indeed, the percentage of unpaired *VSG* reads is not significantly different to that of adenylate cyclases (3.43–7.53%; $p = 0.892$, independent *t* test), which do not form mosaics but are often arranged in tandem arrays[29] (see Source Data file).

Nonetheless, the few mis-mapped reads could still derive from rare mosaic transcripts. To examine these explicitly, we aligned

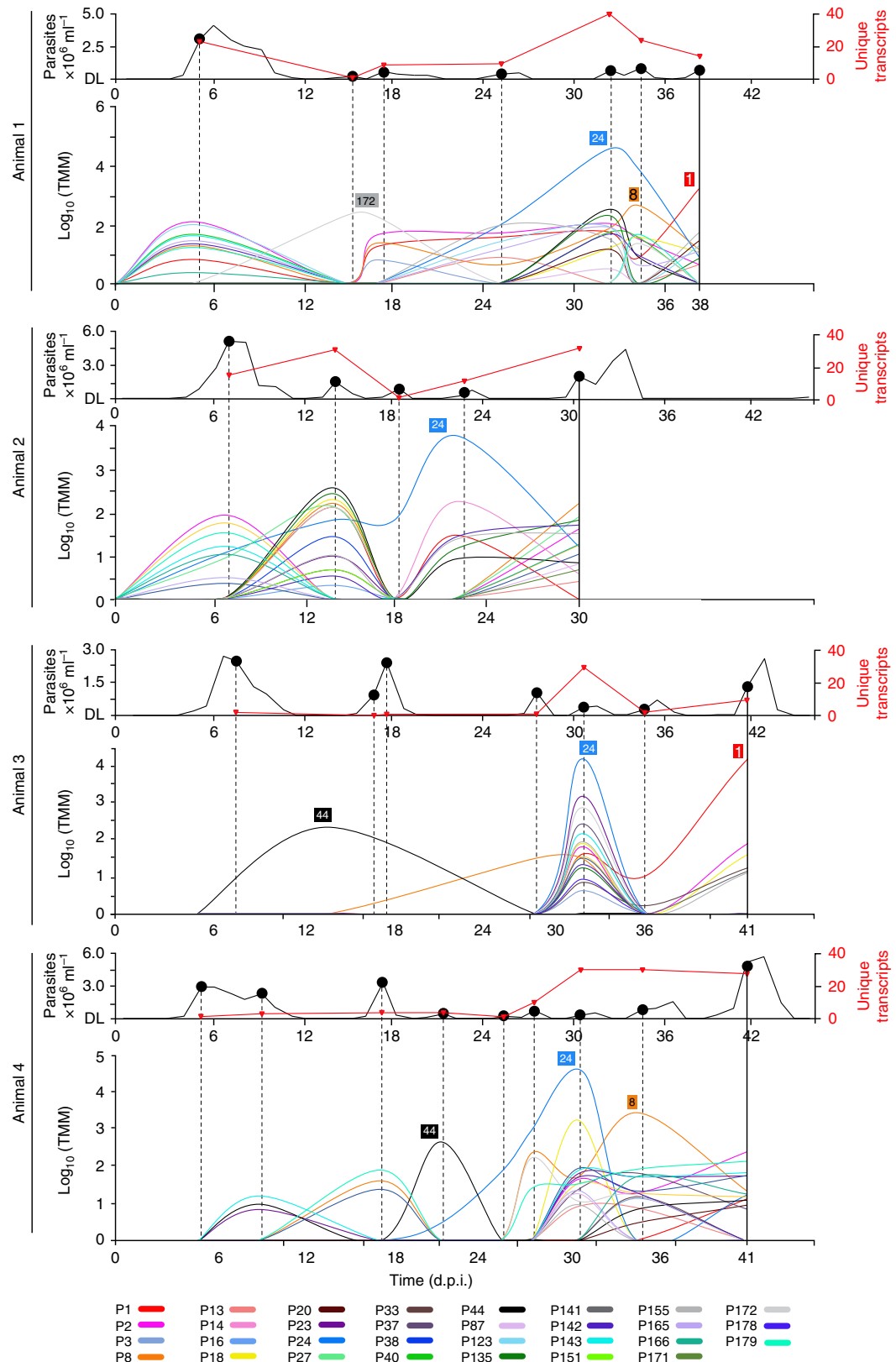

*VSG* transcripts with the three most similar genes from the *T. vivax* Lins genome sequence using BLASTn (where three sequences >500 bp in length could be obtained; $N = 68$) and used GARD[30] to identify potential recombination breakpoints. The closest matches to each transcript were again always from the same phylotype (minimum full-length sequence identity of 86%).

GARD found that 54/68 alignments displayed significant topological incongruence not attributable to rate heterogeneity, indicating $1.94 \pm 1.66$ breakpoints on average (ranging between 0 and 7). At first, this might suggest that mosaicism is widespread within phylotypes; however, this degree of phylogenetic incompatibility was not significantly different to adenylate cyclases (36/

**Fig. 4 *VSG* phylotype expression during experimental *T. vivax* Lins infections in a goat model (*N* = 4).** Parasitaemia (black line) is shown in the upper graph (detection limit (DL)) was $4.1 \times 10^3$ trypanosome per millilitre of blood). Parasite RNA was isolated at peaks in parasitaemia, indicated as black dots. The number of unique *VSG* transcripts (red line) observed in each transcriptome is plotted on the same axis. The lower line graph shows the combined transcript abundance for each *VSG* phylotype (shaded according to key) through the experiment (days post infection) for four replicates animals (A1–A4 from top to bottom). Note that phylotypes can comprise several, distinct transcripts of variable abundance. Across all peaks in all animals, a phylotype was represented by a single transcript in 105/196 observations, (mean = 1.88 ± 1.26 s.d.). However, across the 31 expressed phylotypes, only eight (P3, P13, P14, P16, P38, P141, P151 and P178) occur as single transcripts on every occasion when they were observed. Thus, while a slight majority of phylotypes are represented by only one transcript at a given peak, most phylotypes are present as multiple transcripts at some point. Phylotypes that were dominant (i.e. superabundant) are labelled adjacent to the pertinent lines. A superabundant *VSG* was defined as having an expression level at least ten times that of the next most abundant *VSG*, and this was observed at 15/28 peaks. For example, P24 is 128 times more abundant than P44 at peak 5 in A1, and P1 is 32 times more abundant than P155 at peak 7. The classical expectation of *VSG* expression is that a peak will include a single superabundant *VSG* like this; often, however, several co-dominant *VSG* phylotypes occurred with comparable expression levels, for example at peak 1 in A1 and A2. Source data are provided as a Source Data file.

48 alignments with significant topological incongruence and an average of 1.87 ± 1.88 breakpoints (ranging between 0 and 8); $p = 0.39$, independent *t* test) (see Source Data file).

In summary, while most transcript alignments contained breakpoints, these only implicated very similar sequences, and the scale of genetic admixture was comparable to other tandemly arrayed gene families. Thus, we believe that the apparent assortment of these paralogous sequences is consistent with re-arrangements (real or artefactual) caused by tandem arrangement of *T. vivax VSG*. Certainly, no transcript contained evidence for mosaics of different VSG phylotypes and therefore, assortment of the same magnitude as *T. brucei* was not seen.

## Discussion

In the current model of trypanosome antigenic variation, recombination is the driver behind the creation of novel *VSG* sequences and persistence in the mammalian host. Unlike *T. brucei* and *T. congolense*, we find little evidence for *VSG* mosaics, either historically in the population or during experimental infections. Instead, *T. vivax VSG* repertoire comprises 174 conserved phylotypes, and incomplete sorting of these lineages produces population variation. Although there are *T. vivax* populations in Eastern and Southern Africa that we have yet to sample, we see now that the deep ancestry of *VSG* lineages and lack of *VSG* pseudogenes in *T. vivax*[10] reflect a long history without recombination.

Experiments in the twentieth century documented the progression of variant antigen types (VATs) during *T. vivax* infections[3,16,17]. VATs represent parasite clones that confer a specific, reproducible immunity, assumed to relate to a specific *VSG*. Our results confirm the hypothesis that emerged from these experiments that the *T. vivax VSG* repertoire is smaller than those of other species[3,16]. While the number of *VSG* genes is comparable to *T. brucei* and *T. congolense*, these provide fewer unique antigens because they are often extremely similar, are expressed simultaneously, and cannot recombine. This explains several features of *T. vivax* infections, including the propensity for host self-cure[16] and the re-emergence of VATs late in infection[17]. Furthermore, 70% of phylotypes and 45% of COGs are shared between East and West Africa respectively, which could explain the widespread distribution of serodemes, that is, why immunity to VATs in East Africa provides protection against some parasite strains from Western and Southern Africa also[19,31].

We have defined VSG phylotypes as universal but variable quantities for variant antigen profiling of any *T. vivax* strain. The evolutionary conservation of many phylotypes, and their reproducible expression patterns (in contrast with individual genes), has shown that phylotypes are not merely a convenient means of classifying *T. vivax VSG*, but must have biological relevance. How

they are relevant to the mechanism of antigenic variation in *T. vivax* is not yet clear. Recently, it was suggested that patterns in *T. brucei VSG* expression could relate to protein length, with longer VSG being expressed as infections progress[32]. Our data do not support this hypothesis in *T. vivax*, there was no correlation between VSG length and days post infection, using either our own transcript lengths or the full-length, cognate Y486 gene (see Source Data file). Regardless of why these phylotypes are ubiquitous, a crucial consideration is how they relate to VATs. If individual transcripts in a phylotype cross-react with the same antibody, then VATs are likely to be synonymous with phylotypes, which raises the question of why multiple transcripts are expressed when this confers no benefit to parasite persistence. Conversely, if all *VSG* transcripts represent serologically distinct proteins, this poses the question of why co-expression is determined by sequence homology. Either way, the relevance of VSG phylogeny to antigenic variation is clear.

The absence of recombination among VSG genes means that the generation of antigenic diversity in *T. vivax* differs from *T. brucei*. We propose, based on the current model of *T. brucei* antigenic variation, that the mechanism of VSG switching must also be different. Although it should be stressed that antigenic diversity and antigenic variation are distinct processes, it is important to realise that the two are intricately linked in *T. brucei* by mechanism. *T. brucei* VSGs are expressed from dedicated cassettes at the telomeres of megabase chromosomes called VSG expression sites (ES)[33–35]; note that analogous structures have not yet been found in *T. vivax*. In *T. brucei*, VSG switching can occur through activation of alternative ES by epigenetic means[36–38], or through the substitution of the VSG in the active ES mediated by repetitive DNA motifs that do not occur in *T. vivax*[10]. The latter can occur through SGC, or telomere exchange, and is mostly triggered by double-strand breaks[39–41]. The other consequence of SGC is the creation of mosaic VSG sequences in *T. brucei*, that is, antigenic diversity. Therefore, while we have not examined the mechanism of *T. vivax* antigenic variation directly, we propose that, whatever the mechanism is, it must be different to the *T. brucei* model because *T. vivax* VSG show no evidence of mosaicism historically, or during infections. We do not insist that recombination never happens in *T. vivax*. Among highly similar VSG sequences, for example those found in tandem gene arrays[10], we would expect either allelic or ectopic recombination to occur when these sequences align. Furthermore, meiotic recombination during sexual reproduction (which is unconfirmed in *T. vivax*) could promote gene flow and antigenic diversity among parasite populations. Indeed, recombination could still play a role in *T. vivax* VSG switching, just not in a manner that promotes sequence diversity. We have observed that multiple, related *T. vivax* VSG transcripts are often expressed simultaneously; if these co-expressed transcripts derive from the same tandem array

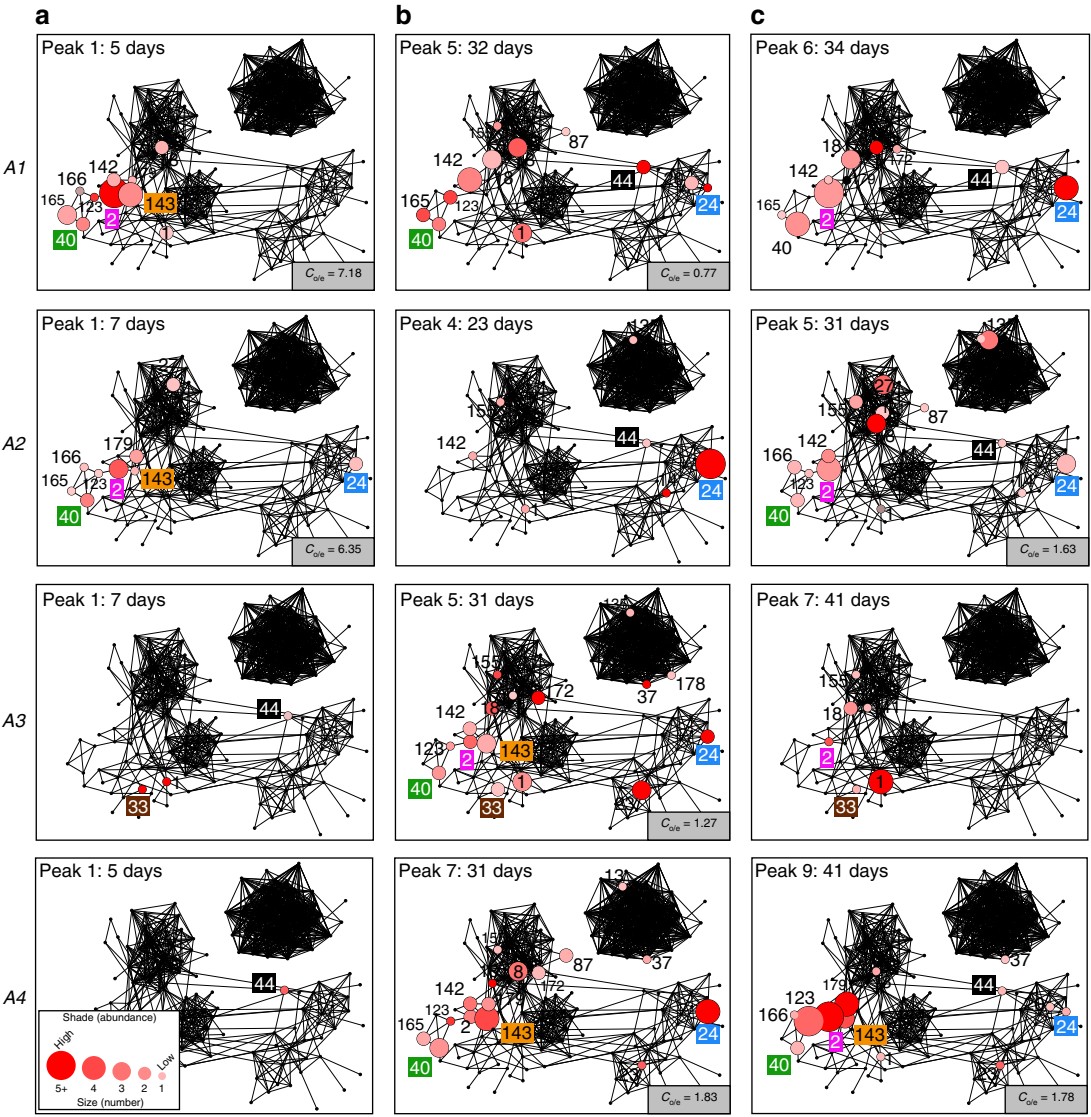

**Fig. 5 Expression of *VSG* phylotypes in the context of sequence similarity.** Combined transcript abundance for expressed phylotypes are plotted on to the phylotype sequence similarity network at **a** early (Peak 1), **b** middle (peaks 4–7), and **c** late (last peak) infection stages respectively. Data from four replicate animals are shown (A1–A4 from top to bottom). Nodes represent phylotypes and are labelled by phylotype number. Node size indicates the number of unique expressed transcripts, while node shade indicates the combined transcript abundance (log$_2$ CPM). The classical expectation of *VSG* expression is that a dominant VSG should subside in abundance and disappear as the host acquires antibody-mediated immunity. However, phylotypes were seen to persist across peaks and/or re-emerge later in the experiment; for instance, P40, P24 and P33 are present at all three time-points in A1, A2 and A3 respectively. Similarly, P2 was expressed strongly at the beginning and re-emerges at the end of infections in A1 and A2. Likewise, P44 was expressed at both the beginning and end of infection in A4. Since only three time-points are shown, it should be noted that these phylotypes were not present at all peaks, so this could represent re-emergence rather than persistence. In cases where sufficient nodes were expressed, the clustering coefficient (C) for their sub-network was calculated. This observed value was compared to mean average C for 100 randomised sub-networks of the same size. The ratio of the observed and expected (by chance) clustering coefficients for expressed sub-networks is shown where a calculation was possible. This value typically exceeds one showing that expressed nodes cluster more than random selections. When considered over all peaks, the clustering coefficient of expressed nodes is significantly higher than coefficients of randomised sub-networks of the same size (see Supplementary Fig. 10 for further details). Source data are provided as a Source Data file.

of VSG paralogues, rearrangement of the arrays could have a central role in a distinct switching mechanism not dependent on SGC.

Without recombination to create mosaic *VSG* sequences, there is a fundamental limitation on antigenic diversity in *T. vivax* and therein its capacity for immune evasion. This poses profound new questions of how *T. vivax* persists long enough to transmit (which it evidently does very successfully). Perhaps *T. vivax* has adopted a different life strategy with respect to the

transmission-virulence or invasion-persistence trade-offs that govern pathogen evolution[42,43]. One possibility is that *T. vivax* has evolved a more acute infection strategy than other species and achieves transmission over shorter periods. Some aspects support an invasion-persistence trade-off; *T. vivax* infections (where the host survives) are typically shorter than other species[44,45], and some haemorrhagic strains cause an extremely acute syndrome that is also hypervirulent[46,47]. Furthermore, where trypanosome species have been directly compared,

chronic pathologies such as reduced packed cell volume[44,45] and humoral immunosuppression[48] are less severe with *T. vivax*. However, there is no evidence that *T. vivax* replicates or transmits quicker, as would be expected under a trade-off. Another possibility is that the idiosyncratic life cycle and wider vector range of *T. vivax*[6] are an adaptation to increase transmission in the absence of long-term persistence in the mammalian host. However, in various reports, animals that survive the initial acute *T. vivax* infection are said to develop a chronic, often asymptomatic, infection during which parasites are not visible[49–51], but which may cause progressive neuropathy[52]. Thus, another possibility is that *T. vivax* cause long-term, chronic infections like other species, but has an alternative mechanism for persistence. Dissemination to immune-privileged sites might allow persistence at low cell densities and *T. vivax* does disseminate to the reproductive and nervous systems, but all trypanosome species have a comparable ability for disease tropism[53].

In conclusion, the orthology of VSG phylotypes across populations, and the considerable structural divergence among them, indicates that the *T. vivax* variant antigen repertoire lacks the dynamism typical of *T. brucei* VSG. Crucially, we find no evidence in *T. vivax* for the vital role that recombination has in diversifying *T. brucei* VSG sequences. This is a major departure from the current model of antigenic variation, indicating that *T. vivax* has a distinct mechanism of immune evasion. Antigenic diversity is limited in *T. vivax*, but not in *T. brucei* and *T. congolense*; this both explains the antigenic exhaustion observed during *T. vivax* infections and poses important new questions of how infections persist under such circumstances. Possibly, the lack of adaptation for persistence reflects a fundamentally different life strategy in *T. vivax*, with profound implications for understanding virulence and transmission of this pervasive and devastating pathogen.

## Methods

**Ethical considerations**. This study was conducted in accordance with the guidelines of the Brazilian College of Animal Experimentation (CONCEA), following the Brazilian law for 'Procedures for the Scientific Use of Animals' (11.794/2008 and decree 6.899/2009). Ethical approval was obtained from the Ethical Committee to the Use of Animals (CEUA) of the Veterinary and Agrarian Sciences Faculty (FCAV) of the State University of São Paulo (Jaboticabal campus) (São Paulo, Brazil) (protocol no. 001494/18, issued on 08/02/2018). The study was also approved by the Animal Welfare and Ethical Review Body (AWERB) of the University of Liverpool (AWC0103).

**Sample preparation**. A panel of 25 *T. vivax*-infected blood stabilates (150 µl), representing isolates from Burkina Faso ($N = 5$), Ivory Coast ($N = 3$), Nigeria ($N = 11$), Gambia ($N = 1$), Uganda ($N = 4$), Togo ($N = 1$), were selected from Azizi Biorepository (http://azizi.ilri.org/repository/) at the International Livestock Research Institute (ILRI), and the Centre International de Recherche-Développement sur l'Elevage en zone Subhumide (CIRDES). In addition, genomic DNA of three Brazilian isolates previously described[24,54,55] was obtained from Instituto de Ciências Biomédicas (ICB) at the University of São Paulo. For samples from ILRI and CIRDES: Red blood cells were lysed with ACK lysing buffer (Gibco, UK) and discarded by centrifugation. Cells were washed twice in 1 ml MACS buffer by centrifugation (10 min, 2500 rpm). The pellet was re-suspended in 100 µl lysis buffer (aqueous solution of 1 M Tris-HCl pH8.0, 0.1 mM NaCl, 10 µM ethylenediaminetetraacetic acid (EDTA), 5% sodium dodecyl sulfate (SDS), 0.14 µM Proteinase K). Samples were incubated at room temperature for 1 h and DNA was extracted with magnetic Sera-Mag Speedbeads (GE Healthcare Life Sciences, UK) according to the manufacturer's protocol. For samples from ICB: DNA was extracted following a traditional phenol-chloroform extraction protocol (TvBrRp) or an ammonium acetate protocol as previously described[56] (TvBrMi). In summary, sample digestion was performed in Digsol buffer (50 mM Tris, 20 mM EDTA, 117 mM NaCl and 1% SDS) with Proteinase K (final concentration 10 mg/ml) for 3 h at 55 °C. DNA was precipitated with ammonium acetate (2.5 M final concentration). DNA purification was performed with ethanol precipitation and 70% ethanol washing. Pellets were air-dried for 30 min, DNA re-suspended in 50 µl sterile water and samples stored at −20 °C until use.

**Genome sequencing and assembly**. Illumina paired-end sequencing libraries were prepared from genomic DNA using the NEBNext® Ultra™ DNA Library Prep Kit according to the manufacturer's protocol (New England Biolabs, UK) and sequenced by standard procedures on the Illumina MiSeq platform, as 150 bp (ILRI) or 250 bp (ICB and CIRDES) paired ends. For each sample, the data yield from sequencing after quality filtering was between $1.69 \times 10^6$ and $1.32 \times 10^7$ read-pairs. Samples were assembled de novo using Velvet 1.2.10[57] with a kmer of 65 (ILRI and CIRDES) or 99 (ICB). These produced assemblies with n50 between 238 and 2852 bp (median = 353; mean = 985). The ratio of read depth of an alternative allele over the total read depth was calculated to detect mixed infections[58,59]. All samples were from single infections only.

**VSG-like sequence recovery and systematics**. *VSG*-like nucleotide sequences were retrieved from the assembled contigs files by sequence similarity search with tBLASTx (v2.8.0)[28]. We used a database of *T. vivax* Y486 *VSG* as query and a significance threshold of $p < 0.001$, contig length ≥100 amino acids, and sequence identity ≥40%. Additionally, we queried a database of *T. brucei* a-*VSG* and b-*VSG* sequences, using the same $p$ value and length thresholds, to accommodate *VSG* genes that might be absent from *T. vivax* Y486, i.e. the possibility that the reference genome is not representative of all strains. In the event, the reference proved to be representative.

*VSG*-like sequences were translated and clustered using OrthoFinder[60] under the default settings. Orthofinder clustered orthologous sequences from the reference and 28 strains. In practice, these clusters of orthologues ('COGs') also included near-identical in-paralogues. Sequences in each cluster were aligned using Clustalx (v2.1)[61] and all alignments were edited to remove overhangs and short (<100 bp) sequences. Edited alignments were refined to produce COGs with >90% average sequence identity by combining COGs that were very similar or, more frequently, subdividing Orthofinder clusters that contained several orthologous groups until the average sequence divergence was <0.05. In complex cases of large Orthofinder clusters, neighbour-joining phylogenies were estimated to aid subdivision. Sequences that could not be placed with any other such that sequence divergence was <0.05 were categorised as 'unclustered' (assumed to be strain-specific *VSG*).

With the membership of COGs determined, we reverted to the original, unedited sequences to identify the longest representative of that COG (a 'type sequence'). Type sequences were combined with the original, unclustered sequences and compared with Fam23–26[13] *VSG* reference sequences using BLASTp (v2.8.0) to confirm their validity and assign a subfamily[10]. We obtained 2039 type sequences in total, 961 belonging to Fam23, 543 to Fam24, 244 to Fam25, and 191 to Fam26. Sequences found not to have a satisfactory match to Fam23–26 *VSG* were excluded. This process produced 760 COGs (comprising 2582 sequences) and 1279 unclustered, or 'singleton' sequences. Each type sequence and singleton was compared against all others using BLASTp to establish cohorts of related COGs/singletons, which we call 'phylotypes'. A BLASTp output was used to create sequence alignments for phylotypes and to estimate neighbour-joining phylogenies for each. The membership of phylotypes was manually adjusted by removing the most divergent sequences until each met a threshold of 70% average sequence identity.

Note that the geographical distribution *VSG* COGs and phylotypes is inferred from the strains in which type sequences were detected. We define a 'cosmopolitan' COG or phylotype as being present in more than one location, except if these locations are Brazil and Uganda, or any combination of Ivory Coast, Togo and Burkina Faso. In both cases, we judged the *T. vivax* strains to be too close to justify these as separate populations. COGs or phylotypes found only in Brazil and Uganda are considered 'East African' in this study. Those found only in some combination of Ivory Coast, Togo and Burkina Faso are considered 'West African'.

**Variant antigen profiling**. To produce VAPs for each strain, we used sequence mapping to confirm the presence or absence of individual COGs. For a gene to be present in a strain, at least one read must map with a threshold nucleotide identity of 98% (allowing a maximum of five nucleotides mismatch per 250 bp read). As mapping makes use of low-coverage reads that would not otherwise be integrated into *VSG* sequence assemblies, this was more efficient than inspecting genome contigs for sequence homology. There was an 11% increase in the observed repertoire size (an average of 87 additional *VSG*) when mapping relative to BLAST. Mapping indicated that most singleton sequences were present in other strains despite the absence of assembled orthologues. Specifically, from the 1279 sequences that could not be placed in a COG with BLAST, only 34 (2.7%) remained unplaced after mapping. For these reasons, trimmed sequence reads were aligned to the 2039 COG type sequences, using Bowtie2 (v2.3.4)[62] set to -D 20 -R 3 -N 1 -L 20. A customised Perl script was used to select entries with a match length ≥245 nucleotides (corresponding to a 2% error rate in a 250 bp sequencing read), mapped as proper pairs, in the correct orientation, and within the expected insert size. This list was compared to the COG database and used to produce the presence/absence binary matrix that represents the *T. vivax* VAP. VAP-based strain relationships were estimated by hierarchical clustering analysis in R, using binary distance calculation and the Ward's minimum variance method[63], and compared to the whole-genome variation phylogeny. For phylotype-based VAPs, presence/absence and distribution data were generated by summing over all constituent *VSG* COGs and singletons.

**Strain variation**. To estimate strain relationships based on the whole genome, MiSeq reads were retrieved and mapped against the *T. vivax* Y486 genome using BWA mem[64], converted to BAM format, sorted and indexed with SAMtools (v1.9)[65]. Sorted BAM files were cleaned, duplicates marked and indexed with Picard (http://broadinstitute.github.io/picard/), and single nucleotide polymorphisms (SNPs) were called and filtered with Genome Analysis Toolkit suite (v3.8-0) according to the best practice protocol for multi-sample variant calling[66]. The multi-sample VCF file obtained from GATK was converted to FASTA format using VCFtools v0.1.14[67] and a maximum likelihood phylogeny was estimated with PHYML (v3.0)[68], using the GTR + Γ + I model of nucleotide substitution, following Smart Model Selection[69].

***T. vivax* experimental infections**. Five male Saanen goats of 4−8 months of age, housed at the Veterinary and Agrarian Sciences Faculty (FCAV) of the State University of São Paulo (Jaboticabal campus) (São Paulo, Brazil), were infected with the *T. vivax* Lins[24] isolate. Before inoculation, parasite stabilates cryopreserved in 8% glycerol were thawed, checked for viability under a light microscope. Each animal was inoculated intravenously with approximately $6 \times 10^6$ parasites. There is no in vitro culture system for bloodstream-stage *T. vivax*. Therefore, it is not currently possible to derive single clones from blood, and the frozen stabilate used here represent mixed populations of the antigenic types circulating the donor animal prior to the experiment. However, we can expect one or two clones to be dominant within these populations and all animals received aliquots of the same preparation from one donor. Animals were clinically examined daily and parasitaemia was determined by microscopy using the Brener's method[70]. This consists of loading 5 μl of blood into a slide with a coverslip, counting the number of trypanosomes present in 50 microscopic fields of view, and multiplying by the microscope correction factor. Animal 2 was euthanised by anaesthesia overdose on day 39 post infection (p.i.) after showing signs of health deterioration (loss of appetite, lethargy and anaemia). Xylasine chlorohydrate (0.2 mg/kg) was administered intra-muscularly as pre-anaesthetic medication, followed by intramuscular ketamine chlorohydrate (2 mg/kg) as anaesthetic. Cardio-respiratory arrest was induced by intrathecal administration of lidocaine chlorohydrate. Remaining animals were euthanised on day 45 p.i. according to the same procedure.

**Blood collection, RNA extraction and sequencing**. At each parasitaemia peak, 4 ml of blood were collected from jugular venepuncture and centrifuged for 15 min at $13,000 \times g$. The buffy coat was removed into a 2.0 ml LoBind microcentrifuge tube (Eppendorf, UK), 1.5 ml of ACK Lysing buffer (Gibco, UK) added, and the mixture incubated for 15 min at room temperature to lyse leftover red blood cells. Samples were centrifuged for 15 min at $13,000 \times g$, washed twice in PBS, pH 8.0, snap-frozen in liquid nitrogen and kept and −80 °C until RNA extraction. RNA was extracted from $2.23 \times 10^5$ to $1.04 \times 10^7$ trypanosomes per sample, using the RNeasy Mini Kit (Qiagen, UK) according to the manufacturer's protocol. RNA yields varied between 117 ng and 13 μg per sample, quantified on the NanoDrop 2000 (ThermoFisher Scientific, Brazil). Up to 1 μg of total RNA was used to prepare multiplexed cDNA libraries as described[71] using the *T. vivax* splice-leader (SL) sequence[72] as the second cDNA strand primer. Briefly, this consists of amplification of first-strand cDNA using a random primer linked to an Illumina adaptor sequence, selection of trypanosome cDNA by amplification of the secondary cDNA strain using a reverse primer containing the SL sequence linked to an Illumina adaptor sequence, and addition of Illumina sequencing adaptors and double-strand cDNA amplification. Finally, Illumina Nextera indexes were added by PCR amplification to allow multiplexing. For samples up to day 30 p.i., this protocol was followed exactly as described, quantified using Qubit HS dsDNA (Invitrogen, UK) and the Agilent 2100 Bioanalyzer (Agilent Technologies, UK), and sequenced at Centre of Genomic Research (Liverpool, UK) on a single lane of the HiSeq 4000 platform (Illumina Inc, USA) as 150 paired ends, producing 280 M mappable reads. However, as the library insert sizes produced were longer than recommended for the HiSeq 4000 platform (Illumina Inc, USA), the protocol for samples from days 30 to 45 p.i. was modified. Instead of adding the indexes from the Illumina Nextera index kit, adapter-ligated, SL-selected cDNA was used as input for the NEB Ultra II FS DNA library kit (NEB, UK), which includes an initial step of DNA fragmentation. Sequencing statistics are shown in Supplementary D1.

**Transcriptome profiling**. RNAseq reads were assembled de novo using Trinity (v2.8.6)[73]. Transcript abundances were estimated for each sample with kallisto (v0.45.0)[74] using Trinity pre-compiled scripts. Subsequently, transcript abundances of samples from the same animal, expressed as transcripts per million, were combined and normalised based on the weighted trimmed mean of log expression ratios (trimmed mean of M values (TMM)[75]. TMM normalisation adjusts expression values to the library size and reduces composition bias. TMM values were used to produce transcript expression matrices for each animal. To recover all *VSG*-like sequences in the transcriptomes, a sequence similarity search was performed with tBLASTx[28] using the *T. vivax* COG database produced above as query and a significance threshold of *E* < 0.001, contig length ≥150 amino acids, and sequence identity ≥70%. All retrieved *VSG*-like sequences were

manually curated to remove spurious matches. The resulting lists of *VSG* transcripts were used as query in a sequence similarity search to identify *VSG* transcripts matching the list of COGs defined in the VAP. A threshold of *E* < 0.001, contig length >50 amino acids, and sequence identity ≥98% was applied. Finally, *VSG* transcripts were assigned a phylotype based on sequence similarity comparison to the VSG phylotype network (≥70% nucleotide identity across the whole gene sequence). *VSG* transcript abundances were combined per phylotype, resulting in a transcript expression matrix containing the abundance of each VSG phylotype over time.

**Recombination analysis**. Fifty previously published genomes from *T. brucei* spp[29,76,77]. and *T. congolense*[20] and 19 of the *T. vivax* genomes presented in this study were used to compare signatures of recombination across species (Supplementary Data 4). *VSGs* and adenylate cyclase genes were extracted from genome assemblies by sequence similarity search (BLASTn[28]) using a nucleotide identity ≥50%, length ≥ 600 nucleotides, and *E* < 0.001. *VSG* assortment was quantified by read mapping using Bowtie2[62]. *VSG* read-pairs were retrieved from the genomes and mapped against reference full-length *VSG* to calculate the proportion of strain read-pairs remaining paired after mapping. This protocol was repeated for adenylate cyclases to provide a negative control; adenylate cyclase genes are multi-copy and include numerous, tandemly arrayed near-perfect copies, but do not undergo SGC[13,29].

In the segmental mapping approach, reference *VSGs* were broken into 150 bp fragments and mapped against the strain *VSGs* to calculate the frequency of reference reads remaining paired. *VSGs* were characterised into UC, MC and FC, according to the estimated number of donors (i.e. the reference VSGs from which the pseudo-reads were derived). FC *VSGs* were those with at least one donor contributing to more than 84% of the sequence. MC *VSGs* were those with one or more donors contributing with more than one fragment (≥300 bp), whereas UC *VSGs* were those remaining (i.e. one or more donors contributing with one fragment only (i.e. ≤150 bp)). The reference *VSGs* that were not mapped at least once to the strain *VSGs* were considered reference-specific variants.

Evidence for past recombination within alignments of MC and FC *VSGs* and adenylate cyclases was examined using phylogenetic incompatibility in PhiPack[23]. The proportion of alignments showing significant phylogenetic incompatibility ($P_{pi}$) was calculated using PhiPack and compared to the $P_{pi}$ of two sets of simulated data (250 replicates, 16 artificial sequences per replicate) with and without recombination. Simulated data were generated with NetRecodon (v6.0.0)[78], under diploid settings, a population mutation rate ($\theta$) of 160, a heterogeneity rate of 0.05, and an expected population size of 1000. The population recombination rate ($\rho$) was set to 0 and 96 for the non-recombinant dataset and recombinant datasets, respectively. Both experimental and simulated sequences were divided into sequence quartets, aligned with Muscle (v3.8.31)[79] and iteratively parsed through PhiPack[23]. MC *VSG* quartets were created with one MC VSG plus three donor VSGs from the reference. FC *VSG*, adenylate cyclase and simulated quartets were randomly generated and parsed through PHI 100 times for statistical power. MC quartets were compiled manually with MC VSG and three donors.

Total sequence orthology in each trypanosome species *VSG* repertoire was calculated as the proportion of total *VSG* nucleotides of a strain repertoire shared with the relevant reference genome, averaged over all strains. This includes all FC *VSG* and conserved segments of MC *VSG*. The number of shared nucleotides was extracted from the mapping output file using genomecov from BEDtools (v2.27.0)[80].

**Estimation of ancestral recombination graphs**. Ancestral recombination graphs were reconstructed for multi-coupled and fully coupled *VSG* quartet alignments and adenylate cyclase control quartet alignments using the ACG software package[81]. The TMRCA was estimated along the length of each aligned quartet at 20 bp intervals using a 100-bp-wide sliding window using constant recombination rate/ population size models with an MCMC length of 10,000,000, burn-in of 1,000,000 and sampling frequency of 2500. For each individual quartet the TMRCA along the length of the alignment was summarised by calculating the mean TMRCA. To identify evidence of recombination, which would generate a sequence with regions of differing ancestries, the variance in TMRCA along the alignment was calculated for each individual quartet.

**Reporting summary**. Further information on research design is available in the Nature Research Reporting Summary linked to this article.

## Data availability

The datasets generated during the current study are available in the NCBI repository, under the ENA project accession number PRJNA486085. The source data underlying Figs. 1–3a, 3c–5 and Supplementary Figs. 1 and 8−10 are provided as a Source Data file.

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

## Acknowledgements

This work was supported by grants from the Biotechnology and Biological Sciences Research Council (BB/M022811/1 and BB/R021139/1), an International Veterinary Vaccinology Network (IVVN) pump-priming award, a Bill and Melinda Gates Foundation Grand Challenges Explorations award (Round 11), and the Wellcome Trust (WT206815/Z/17/Z).

## Author contributions

Conceived and designed the experiments: S.S.P., A.P.J. Performed the experiments: S.S.P., H.N., M.O., K.J.G.d.A.C.N. Analysed the data: S.S.P., C.W.D., P.R., A.P.J. Contributed reagents/materials/analysis tools: M.R.A., Z.B., S.K., R.Z.M., M.M.G.T., A.P.J. Wrote the paper: S.S.P., A.P.J. Obtained funding: S.S.P., M.M.G.T., R.Z.M., A.P.J.

## Competing interests

The authors declare no competing interests.
