## [Peer Review File · Nature Communications]

Reviewers' Comments:

Reviewer #1:

Remarks to the Author:

The work of Silva Pereira et al. combined genome and transcriptome sequencing to test the hypothesis that VSG diversity in *T. vivax* is not driven by recombination. Earlier work by Jackson et al. (2012) on single reference strains has already shown that recombination is more frequent among *T. brucei* and *T. congolense* VSG than in *T. vivax*. However, here they present the first population-based identification of VSG in *T. vivax*, demonstrating that the VSG repertoire is broadly conserved and follows the *T. vivax* evolutionary history. This congruent signal is very different to earlier observations in *T. congolense* where a similar population-based VSG examination revealed a clear discordance between nuclear and VSG ancestries (Silva Pereira et al., 2018). In addition, the authors performed experimental animal infections in conjunction with transcriptomics, and show reproducible expression patterns of VSG phylotypes. While the biological relevance of this observation is not immediately obvious, the experiments present very interesting observations that could drive future studies on the role of co-expression of related, non-identical transcripts to the progression of *T. vivax* infections.

Based on the data presented, I'm convinced that recombination does not play a major role in explaining *T. vivax* VSG diversity at the phylotype level. Using a high standard of bio-informatic analyses, the authors show convincingly that VSG recombination is lower in *T. vivax* compared *T. congolense* / *T. brucei*. Their results support their main conclusion that the mechanism of VSG switching in *T. vivax* must be different to the *T. brucei* model.

However, the fact that VSG recombination is lower in *T. vivax* compared to the other trypanosome species does not imply that recombination is absent. For instance, while Figure 3a and 3c show a significant difference between trypanosome species, there is also a considerable overlap of the data ranges. Is this due to differences in coverage, or could it indicate that some *T. vivax* isolates show evidence of VSG recombination? Related to this, Figure 1 shows discordant patterns in the West African clade and these isolates show VSG profiles that seem to be present in the East African clade and the Nigeria clade. This observation was not discussed in the paper. Is it possible that the West African isolates show evidence of VSG recombination? If yes, could this perhaps suggest that *T. vivax* can be split into two groups: one showing no evidence of VSG recombination, and the other showing some evidence of *T. brucei*-like VSG recombination?

Another point of discussion is the type of recombination. Ectopic recombination is the most likely mechanisms of VSG switching, but the observations here show a striking difference between the clonal *T. vivax* and the more sexually active *T. congolense* and *T. brucei*. Could this imply that - beside ectopic recombination - the rate of meiotic recombination could also be important in shaping VSG diversity globally?

Some suggestions of minor improvements:

I'm a bit puzzled about the huge variation in coverage statistics. I think it would be good to add some figure or statistics to show that the poor coverage didn't affect downstream analyses. My guess is that the poor coverage is the main reason why the authors had to resort to the VSG phylotyping for individual profiling (line 122-127)?

Line 241: Diverged by up to 26.5%. I'm not sure how to interpret this number.

Line 314: Grammar failure.

Line 421: Do you mean the frequency of allelic read depths? Perhaps worth citing a paper that describes in more detail how it could be used to determine mixed infections in trypanosomes.

Line 444: How many COGs were determined?

Line 448-450: Confusing section. What exactly are the numbers between brackets on line 448? These numbers sum to 2044, which is not the same as the 2576 sequences that were assigned to a COG. I assume it is also not the number of type sequences.

Line 449: Did you really exclude the sequences with an unsatisfactory match? I thought that these were included as singleton sequences.

Line 457: There is a word missing in this sentence.

Line 472: I don't understand what is being said with the sentence "Of 1279 sequences only 2.7% remained location-specific".

Line 476-477: What was the minimum number of read-pairs and nucleotide coverage needed to regard a COG as present?

Lines 537 and 540. $E > 0.001$?

Line 560: Define donors.

Line 569-571: It is unclear how these simulations were parametrised. How were the parameters chosen? The authors used a similar population size for the three species, but it is imaginable that the population size of *T. vivax* is considerably smaller than that of *T. congolense* and *T. brucei*. Could such differences in population history explain the lower VSG recombination of *T. vivax* compared to the other two trypanosome species?

Line 572: How were quartets exactly defined? For MC quartets, I guess you used the MC VSG of the strain together with 3 donor VSGs of the reference? More information is needed here.

Figure 1. The colors are not discriminatory enough. It is hard to disentangle the different countries. Perhaps use different colors or use country names within labels.

Figure 2b - line 796. The way it is represented in Figure 2b suggests that a multi-coupled VSG occurs when reference pseudo-reads from different donors (i.e. different reference VSG genes) map to the same strain VSG. Is my understanding correct? If yes, then the figure legend doesn't make this explicit, as it states that multi-coupled VSG's occurs when reference pseudo-reads map to multiple locations (i.e. different strain VSGs), which is quite different. Also, in the methods line 561, it is stated that multi-coupled VSGs occurs when one or more donors contribute more than 1 fragment. Should this not be at least 2 donors?

Silva Pereira et al. (2018). Variant antigen repertoires in *Trypanosoma congolense* populations and experimental infections extracted from deep sequence data using universal protein motifs. *Genome Research*.

Jackson et al. (2012) Antigenic diversity is generated by distinct evolutionary mechanisms in African trypanosome species. *PNAS*. 109: 3416-3421

Duffy CW, et al. (2009) Trypanosoma vivax displays a clonal population structure. Int J Parasitol 39:1475–1483.

Reviewer #2:

Remarks to the Author:

In this paper, Pereira et al describe how in *T. vivax*, and unlike in *T. b. brucei* and *T. congolense*, recombination is not the dominant mechanisms through which this parasite has diversified their VSG gene repertoire. Although it had already been shown that recombination is seldom observed in *T. vivax* (PMID:22331916), this paper has gone further to show that both in clinical strains and in experimental infections this holds true. Overall, I think that this is an important piece of work that challenges our understanding of antigenic variation and immune evasion in these parasites. We place a lot of emphasis on recombination as the driver for both VSG gene diversity and as an immune evasion mechanism, but the majority of the work relies on *T.b.brucei* as a model for all trypanosomes. The use, in this paper, of 28 *T. vivax* clinical strains from a broad geographically range reveals just how constrained the VSG family is in these strains.

It is my view that this paper is deserving of publication, however I do have a few comments.

1. There is evidence provided in this paper for a lack of recombination in expanding VSG diversity, strikingly there are a lack of mosaics, but I do not think that there is evidence yet for a complete lack of recombination-based VSG switching. These are two separate issue which should not be conflated. In *T. brucei* there are multiple different sequences that all contribute to VSG switching, including the CTD, 70 bp repeats and the telomere. It is difficult to comprehend how VSG switching could take place in *T. vivax* without any reliance on recombination. To my knowledge no *T. vivax* expression site has been described and whether or not additional repetitive sequences exist that could facilitate recombination are unknow. Therefore there are several statements (line 345/6 and line 377 for example) that, I feel, need to be adjusted.

2. The authors describe how there is a reproducible pattern of expression in terms of phylotypes and some appear to dominate in an infection (p24 and p44 Fig4)and at line 340, the authors state that the phylotypes have 'biological relevance' Can the authors speculate as to why this is seen? Is there anything specific about these phylotypes that would result in them being use at specific points during an infection? It's been recently published in *T. brucei* that that shorter VSG's dominate during and early infection in *T. brucei* (PMID: 30026531). Do the VSG genes in these phylotypes conform to any specific structure?

Minor comment:

3. The authors use 'P' and 'A' from lines 225 and 236. I presume these refer to 'phylotype' and 'animal', but could this be defined when it is first used.

Reviewer #3:

Remarks to the Author:

This paper is about an important livestock pathogen *Trypanosoma vivax* and the mechanism it uses for antigenic variation. On both counts, this is of interest to the wider audience interested in infection and immunity, but as it stands the MS is written for specialists, like the authors, who are very familiar

with the literature on antigenic variation in trypanosomes. Some rewriting is needed to make it accessible to a wider audience. Overall, there appears to be an important finding that antigenic variation in *T. vivax* differs from that in *T. brucei*, the textbook example of antigenic variation. In their genomic survey of 28 *T. vivax* strains, the authors found no evidence of recombined antigen genes (VSG genes). This contrasts with *T. brucei*, where many VSG genes are pseudogenes that need to recombine with other VSG genes in order to be expressed. Thus recombination is an important route to VSG repertoire diversification and expansion in *T. brucei*, but not it seems in *T. vivax*. Secondly, antigen switching in *T. brucei* requires the actively expressed VSG to be replaced in situ by another VSG gene, either partially or wholly. The authors seem to be suggesting that antigen switching works by a different mechanism in *T. vivax*, but need to draw the distinction between these two aspects of antigenic variation more clearly, as well as the distinction between general recombination and the special case of gene conversion - the non-specialist reader may well wonder how gene conversion can lead to greater gene diversity (abstract lines 28,29). Overall, the evidence needs to be presented in a more accurate and lucid way to aid comprehension of both the specialist and non-specialist reader. Detailed suggestions for improvement of the MS follow.

1. The authors have examined 28 *T. vivax* strains mostly from West African countries, but also including some Ugandan and Brazilian strains. While this significantly adds to current knowledge, which is mostly based on analysis of a single lab strain, the authors should acknowledge that their sampling is not comprehensive and does not cover the known range of *T. vivax* diversity. It has long been observed by clinicians that *T. vivax* is more pathogenic in West Africa than East Africa, but here most of East Africa is not represented, only Uganda. Immunological studies have shown limited cross-protection between different *T. vivax* strains in East Africa and phylogenetic analyses have revealed genetic diversity among E African *T. vivax*. It is therefore important to point out to the reader that the conclusions may not apply to the whole of *T. vivax*, because of limited sampling of E African *T. vivax*. Statements such as "continent-wide" (line 158), "immunity to VATs in East Africa" (line 334), "the global *T. vivax* variant antigen repertoire" (line 375, 777) need to be modified accordingly.

2. There are some places where clarity needs to be improved:

Line 34 "either *T. vivax* has an alternate mechanism for immune evasion..." – additional surely? *T. vivax* might have fewer antigenic variants, but nevertheless uses AV for immune evasion. Similar lines 378-9.

Line 35 "Long-term persistence" – in the mammalian host?

Line 52 "truncated life cycle" – implies *T. vivax* life cycle is incomplete. It just achieves its complete life cycle without a stage in the insect gut.

Line 93 "the reference genome repertoire" – which one?

Line 111 "and therefore that VAP reflects both population history and location" – this doesn't follow from the preceding statement that SNP and VAP relationships match without explaining that SNP genotypes are associated with geographical location.

Line 146 a COG is defined as a cluster of orthologs based on $\geq 90\%$ sequence identity, so "if they were relatively recent gene duplications" doesn't make sense, unless each COG comprises 2 genes.

Line 178-180 needs to be rewritten for clarity, defining the meaning of multi-, un- and fully coupled.

Line 240-241 "P24 is a dominant variant antigen" suggests a single entity, but next "the actual P24 transcript expressed was different in each case....diverged up to 26.5%", so there were multiple different P24 genes being expressed. Is there any evidence whether or not the antigens encoded are distinguishable immunologically? This is part of a larger problem for the non-specialist reader, as several different terms are used for grouping VSG genes - VAP, COG, phylotype, Fam – perhaps a Venn diagram to clarify this for the reader?

Line 247-8 delete one "only".

Line 256-7 "more centrally placed" - means mid infection?

Line 271-2 "phylogeny (sequence identity)" – these are not equivalent terms. Do you mean sequence

relatedness? "VSG expression profile" = pattern of VSG expression over time?

Lines 289, 293, 309, 314 "percentage of read-pairs that mapped to unpaired genomic positions", "the percentage of VSG read-pairs split after mapping", "these only implicated very closely related sequences", "therefore, assortment of *T. brucei* order was sort seen" – sentences of uncertain meaning that need rewriting for clarity.

Line 318 "The current model of trypanosome antigenic variation has recombination as the driver behind novelty and persistence" – clarify for the reader, e.g. novelty = the creation of novel VSG genes? Persistence = persistence of infection in the mammalian host?

Line 346-7 The reader needs more information here - what is the mechanism of VSG switching in *T. brucei*? What is known about expression sites in *T. vivax*?

Line 362-3 What *T. vivax* strains were used here? The assumption is that these workers used genetically similar strains of *T. vivax*, but this is not necessarily so.

Line 380 "Antigenic diversity in *T. vivax* is finite, in a way that *T. brucei* and *T. congolense* are not" - doesn't make sense grammatically.

Line 382 "the lack of adaptation for persistence, so evident in *T. brucei*" – this seems to state the opposite of what was intended.

Line 446 need a reference for Fam23-26.

Line 496 "infected with the *T. vivax* Lins isolate." Is *T. vivax* Lins a clone? If not, this significantly alters interpretation of the animal infection data.

Line 499 Since Fig 4 graphs indicate that parasitaemia drops to zero, it is of interest how parasitaemia was determined. Details should be included, as the reference given is obscure (thesis?).

Line 508 It would be useful to mention the estimated number of trypanosomes from which RNA was derived.

Fig 1. Legend - What is meant by "population history"? Figure – it is not obvious what the grey background shading on the left indicates. The colours for text are not readily distinguishable. Could use a 2 letter country code in the strain name instead?

Fig 2. Legend – "global *T. vivax* repertoire" is misleading. Some of the colours are not readily distinguishable, e.g. for cosmopolitan and Nigeria.

Fig 4. Line 829 "classical expectation" – surely the idea that each parasitaemic peak represents a single VSG was realised to be simplistic several decades ago? Indicate on the figure that the graphs pertain to animals 1-4. The parasitaemia is shown decreasing to zero, but there is likely a cut-off of parasite detection above zero. The graphs of detected phylotype are shown as curves, but the RNA data derive only from the peaks of parasitaemia, so what's the justification for showing the data as curves?

Reviewer #1:

1. *However, the fact that VSG recombination is lower in *T. vivax* compared to the other trypanosome species does not imply that recombination is absent. For instance, while Figure 3a and 3c show a significant difference between trypanosome species, there is also a considerable overlap of the data ranges. Is this due to differences in coverage, or could it indicate that some *T. vivax* isolates show evidence of VSG recombination? Related to this, Figure 1 shows discordant patterns in the West African clade and these isolates show VSG profiles that seem to be present in the East African clade and the Nigeria clade. This observation was not discussed in the paper. Is it possible that the West African isolates show evidence of VSG recombination? If yes, could this perhaps suggest that *T. vivax* can be split into two groups: one showing no evidence of VSG recombination, and the other showing some evidence of *T. brucei*-like VSG recombination?*

We do not say that recombination is absent entirely; rather that it does not drive antigenic diversity, as in other species. We suggest that this has consequences for the mechanism of VSG switching, but we do not rule out a role for recombination in antigenic variation. It is still possible that recombination plays a part in switching without assorting coding sequences, we have modified the discussion to make this clear (p16-17, ll396-419). We interpret the presence of COGs and phylotypes across geographically widespread strains to mean that all strains have inherited an orthologous repertoire from their common ancestor. While there are clearly some variations that introduce discordance into Fig1, there is nothing to suggest that West African repertoires buck this general trend; 94% of West African phylotypes are found in strains in other regions. We have altered the text to recognise the discordance in Fig. 1 and suggest reasons for it (p5, ll120-122).

2. *Another point of discussion is the type of recombination. Ectopic recombination is the most likely mechanisms of VSG switching, but the observations here show a striking difference between the clonal *T. vivax* and the more sexually active *T. congolense* and *T. brucei*. Could this imply that - beside ectopic recombination - the rate of meiotic recombination could also be important in shaping VSG diversity globally?*

Yes, this could be implied. Ectopic recombination is thought to be the mechanism of creating diversity among VSG but meiotic recombination would be responsible for new VSG variants moving through the population and spreading between locations. Our view is that *T. vivax* lacks ectopic recombination among VSG, it may also lack meiotic recombination but our present data do not permit a definitive view on this. If present, meiotic recombination could be one way that recombination continues to diversify antigens within populations in *T. vivax*, and we now note this in the discussion (p17, ll412-4).

3. *I'm a bit puzzled about the huge variation in coverage statistics. I think it would be good to add some figure or statistics to show that the poor coverage didn't affect downstream analyses. My guess is that the poor coverage is the main reason why the authors had to resort to the VSG phylotyping for individual profiling (line 122-127)?*

It is a reality that re-sequencing clinical strains with short-read technology will not recover all VSG sequences and we have devised a profiling method that can deal with this. Coverage of the reference genome, shown in Supplementary Table 1, helps us understand how partial the genomes are. In 60% of the strains sequenced, coverage is $\geq 70\%$, and in only 4 instances is this below 50% (including 3 Burkina Faso samples). The reviewer is correct in suggesting that sequence incompleteness is the main reason why we moved from COGs to Phylotypes, the latter being able to accommodate all *T. vivax* VSG we observed. In fact, we explain that COGs cannot be a reliable basis to profiling from the beginning of the manuscript (p6, ll134-145). However, we are convinced that the differences in coverage do not affect our main conclusions because when we take the complete reference genome

(Y486) as an example, we see that while there are strain-specific COGs (p5, l112), (some of which may be falsely missing from other strains due to incompleteness), none of these constitute strain-specific phylotypes. Therefore, even if all our genomes were fully sequenced, we would be unlikely to find many more strain-specific phylotypes. It seems unlikely that the only four genomes with a poor Y486 sequence coverage would all be missing the same set of specific VSGs and that these VSGs were more than 30% different from every other VSG we sampled, many from well-covered genomes.

Therefore, we are confident that despite the VSG complements of our clinical strain genomes undoubtedly being incomplete, (something all studies re-sequencing clinical strains will contend with), we have adopted a profiling approach that is not sensitive to missing data at the gene level. We have sampled enough strains such that, collectively, they reveal the total species repertoire at the phylotype level.

4. *Line 241: Diverged by up to 26.5%. I'm not sure how to interpret this number.*

We mean that the various transcripts belonging to P24 had a minimum of 73.5% nucleotide sequence identity (up to 26.5% divergence). We have updated the text to be clearer (p11, l269).

5. *Line 314: Grammar failure.*

We have corrected this.

6. *Line 421: Do you mean the frequency of allelic read depths? Perhaps worth citing a paper that describes in more detail how it could be used to determine mixed infections in trypanosomes.*

Yes, we do. We have added the reference (p19, ll491-2; refs57,58).

7. *Line 444: How many COGs were determined?*

We determined 2039 COGs (described in Supplementary Table 2).

8. *Line 448-450: Confusing section. What exactly are the numbers between brackets on line 448? These numbers sum to 2044, which is not the same as the 2576 sequences that were assigned to a COG. I assume it is also not the number of type sequences.*

These numbers represented the number of type sequences/COGs. By mistake, we had used outdated numbers. We have now updated them to match the final results analysis, also shown in Supplementary Table 2, and they now add up to 2039 (961 belonging to Fam23, 543 to Fam24, 244 to Fam25, and 191 to Fam26). We have also rephrased the sentence to make it clearer to the reader (p21, ll518-523).

9. *Line 449: Did you really exclude the sequences with an unsatisfactory match? I thought that these were included as singleton sequences.*

We excluded sequences that did not have a satisfactory match to Fam23-26 VSG, i.e. there was no structural homology to justify their inclusion as 'VSG'. If they did have a satisfactory match to these families, but did not match any VSG from the database to more than 90% sequence identity (i.e. they did not fall within a COG), they were included as singleton VSG.

10. *Line 457: There is a word missing in this sentence.*

Apologies, but we cannot see a missing word.

11. Line 472: *I don't understand what is being said with the sentence "Of 1279 sequences only 2.7% remained location-specific".*

We first used BLAST and assembled strain genomes to cluster VSG sequences into COGs. However, as we were concerned that due to low sequencing depth some VSG reads would not have been included into the assemblies, and so would have been missed in the analysis. To overcome this, we used read mapping rather than BLAST to understand the geographical distribution of each VSG sequence. Essentially, it is a reflection of how much better mapping did than BLAST at recording presence that the proportion of COGs found in just one location fell so much. However, the phrasing is confusing and we have changed it (p22, ll547-9).

12. Line 476-477: *What was the minimum number of read-pairs and nucleotide coverage needed to regard a COG as present?*

As we have a binary classification (presence or absence), we did not apply a minimum read-pair number or sequence depth (i.e. 1X was sufficient). A gene was recorded as 'present' if a read matched any 250bp segment at a nucleotide identity threshold of 98% (allowing a maximum of 5 nucleotides mismatch per 250bp read). This is now stated in the methods (p22, ll541-3).

13. Lines 537 and 540. *$E > 0.001$?*

It should be $E < 0.001$. We have corrected this (p25, ll622 and 625).

14. Line 560: *Define donors.*

We have included this (p25, l645).

15. Line 569-571: *It is unclear how these simulations were parametrised. How were the parameters chosen? The authors used a similar population size for the three species, but it is imaginable that the population size of *T. vivax* is considerably smaller than that of *T. congolense* and *T. brucei*. Could such differences in population history explain the lower VSG recombination of *T. vivax* compared to the other two trypanosome species?*

The population size was pre-defined for the simulated data only, not for the VSG sequences. We simulated two sets of artificial sequences, one for population recombination rate of 0 (which is our negative control), and one for population recombination rate of 96 (positive control). For both of these, the population size was set equally for 1000. For the VSG analysis, we did not do any simulations, but rather used the original VSG sequences. Therefore, there were no parameters to define.

Regarding the effect of population history in the VSG recombination, unlike *T. brucei* and *T. congolense*, *T. vivax* may be clonal, which would result in a smaller effective population size, but this is not known for certain. However, as VSG are hemizygous, meiosis would not drive antigenic diversity within VSG genes even if active, although it could change antigen genotype frequencies at a population level. There is no obvious reason why clonality would preclude ectopic or mitotic recombination among VSG, indeed, this seems to happen in the clonal *T. b. gambiense* (several strains of which we use in our *T. brucei* analyses in Fig. 3), and so we do not believe it can explain the negligible signature of VSG recombination.

16. Line 572: *How were quartets exactly defined? For MC quartets, I guess you used the MC VSG of the strain together with 3 donor VSGs of the reference? More information is needed here.*

For MC quartets, that is correct. For FC quartets, we took each FC VSG and grouped it with three other randomly selected FC VSGs. We have included this information in the manuscript (p26, ll659-660).

17. Figure 1. *The colors are not discriminatory enough. It is hard to disentangle the different countries. Perhaps use different colors or use country names within labels.*

We have included country labels and changed the colour codes in Fig. 1 as suggested.

18. Figure 3b - line 796. *The way it is represented in Figure 2b suggests that a multi-coupled VSG occurs when reference pseudo-reads from different donors (i.e. different reference VSG genes) map to the same strain VSG. Is my understanding correct? If yes, then the figure legend doesn't make this explicit, as it states that multi-coupled VSG's occurs when reference pseudo-reads map to multiple locations (i.e. different strain VSGs), which is quite different. Also, in the methods line 561, it is stated that multi-coupled VSGs occurs when one or more donors contribute more than 1 fragment. Should this not be at least 2 donors?*

The understanding about the figure is correct. As such, we have altered the figure legend to be more explicit (p43, ll1088-90). Regarding the methods, it is one or more donors contributing more than one fragment because if we have a strain VSG receiving two fragments from the same reference VSG, we consider it a MC VSG. We define UC VSG only if there is a single reference VSG contributing with a single fragment, or multiple reference VSGs contributing to the same single fragment (in cases where there are gene duplications in the reference genome).

Reviewer #2:

1. *There is evidence provided in this paper for a lack of recombination in expanding VSG diversity, strikingly there are a lack of mosaics, but I do not think that there is evidence yet for a complete lack of recombination-based VSG switching. These are two separate issue which should not be conflated. In *T. brucei* there are multiple different sequences that all contribute to VSG switching, including the CTD, 70 bp repeats and the telomere. It is difficult to comprehend how VSG switching could take place in *T. vivax* without any reliance on recombination. To my knowledge no *T. vivax* expression site has been described and whether or not additional repetitive sequences exist that could facilitate recombination are unknown. Therefore there are several statements (line 345/6 and line 377 for example) that, I feel, need to be adjusted.*

We agree that these issues of antigenic diversity and antigenic variation should not be conflated and we have been careful not to do this. Hence, the title concerns diversity relative to other species, not mechanism. We do not know how VSG switch in *T. vivax*, but the reviewer is correct that no expression site has ever been observed. We know from our previous work that the motifs, named by the reviewer, responsible for switching in *T. brucei* are absent from *T. vivax*. We believe that the lack of recombination among and within VSG coding sequences further suggest that *T. vivax* has a different mechanism to *T. brucei*, if only because, in *T. brucei*, diversity and mechanism are intrinsically linked by a common structural basis (i.e. the motifs mentioned above). Perhaps they are not intrinsically linked in *T. vivax*. Our result does not rule out recombination playing a role in that different mechanism of antigenic variation. We did not discuss this previously in part because we do not wish to conflate our observations of antigenic diversity towards antigenic variation. However, we have re-written the discussion to be clear on this issue (p16-17, ll396-419).

2. *The authors describe how there is a reproducible pattern of expression in terms of phylotypes and some appear to dominate in an infection (p24 and p44 Fig4) and at line 340, the authors state that the phylotypes have 'biological relevance' Can the authors speculate as to why this is seen? Is there anything specific about these phylotypes that would result in them being used at specific points during an infection? It's been recently published in *T. brucei* that shorter VSG's dominate during and early infection in *T. brucei* (PMID: 30026531). Do the VSG genes in these phylotypes conform to any specific structure?*

By 'biological relevance' we mean that the phylotypes are not simply our own creations, or simple systematic devices to categorise VSG. The fact that the same group of related but non-identical VSG occur at similar points in each replicate means that the relatedness upon which our systematics is based has some importance to real infections. We do not yet know how. We thank the reviewer for posing the protein length question, it is very relevant. Having looked at the data, we see no such pattern; based on our own transcripts or, since they are often partial, their cognate genes in the Y486 reference, there is no progression towards longer VSG as the experiment progresses. We have added this observation to the discussion (p15, ll372-377).

3. *The authors use 'P' and 'A' from lines 225 and 236. I presume these refer to 'phylotype' and 'animal', but could this be defined when it is first used.*

Thank you. We have resolved this (p10, l249 and p11, l264).

Reviewer #3:

1. *As it stands the MS is written for specialists, like the authors, who are very familiar with the literature on antigenic variation in trypanosomes. Some rewriting is needed to make it accessible to a wider audience.*

We hope that the specific adjustments made below, and the clarification on terminology in a new Supplementary Figure 2 will satisfy this.

2. *The authors seem to be suggesting that antigen switching works by a different mechanism in *T. vivax*, but need to draw the distinction between these two aspects of antigenic variation more clearly, as well as the distinction between general recombination and the special case of gene conversion - the non-specialist reader may well wonder how gene conversion can lead to greater gene diversity (abstract lines 28,29).*

We agree with this point, as shared by other reviewers. As stated above in response to #1.1 and #2.1, we are suggesting that a different mechanism operates based on the fact that antigenic diversity and the mechanism of VSG switching are intrinsically linked in *T. brucei* by the same process of biased gene conversion. Without any evidence for recombination driving sequence diversity in *T. vivax*, this implies a different mechanism, without ruling out a role for recombination in VSG switching. We have re-written the discussion to acknowledge this (p16-17, ll396-41).

3. *The authors have examined 28 *T. vivax* strains mostly from West African countries, but also including some Ugandan and Brazilian strains. While this significantly adds to current knowledge, which is mostly based on analysis of a single lab strain, the authors should acknowledge that their sampling is not comprehensive and does not cover the known range of *T. vivax* diversity. It has long been observed by clinicians that *T. vivax* is more pathogenic in West Africa than East Africa, but here most of East Africa is not represented, only Uganda. Immunological studies have shown*

limited cross-protection between different T. vivax strains in East Africa and phylogenetic analyses have revealed genetic diversity among E African T. vivax. It is therefore important to point out to the reader that the conclusions may not apply to the whole of T. vivax, because of limited sampling of E African T. vivax. Statements such as “continent-wide” (line 158), “immunity to VATs in East Africa” (line 334), “the global T. vivax variant antigen repertoire” (line 375, 777) need to be modified accordingly.

The reviewer is quite correct that there are locations from which we have not sampled. Having sampled Uganda and Brazil (which are descended recently from East African strains), we believe that our conclusions do extend beyond West African *T. vivax*, but the criticism is valid and we have added a line to the discussion explaining what further sampling would be needed to put these conclusions beyond doubt. We have removed ‘continent-wide’ and replaced this with ‘over large distances’ (p7, l145) and ‘widespread’ (p8, l178). The word ‘global’ has been removed throughout and replaced with ‘total’ (i.e. within our sample). The observation that there west African VATs can give ‘immunity to VATs in East Africa’ is not our observation, but published.

4. *Line 34 “either T. vivax has an alternate mechanism for immune evasion...” – additional surely? T. vivax might have fewer antigenic variants, but nevertheless uses AV for immune evasion. Similar lines 378-9.*

Yes, we have corrected both occasions (p2, l39 and p18, l437).

5. *Line 35 “Long-term persistence” – in the mammalian host?*

We have specified this, thank you (p2, l40).

6. *Line 52 “truncated life cycle” – implies T. vivax life cycle is incomplete. It just achieves its complete life cycle without a stage in the insect gut.*

We have replaced “truncated” for “simpler” (p3, l58).

7. *Line 93 “the reference genome repertoire” – which one?*

We refer to the Y486 strain – we have included this information (p5, l100).

8. *Line 111 “and therefore that VAP reflects both population history and location” – this doesn’t follow from the preceding statement that SNP and VAP relationships match without explaining that SNP genotypes are associated with geographical location.*

That is correct. We have rephrased this section to clarify this. “Fig. 1 shows that strain genealogy estimated from whole genome single nucleotide polymorphisms (SNPs) recapitulates geography and matches the relationships inferred from the VAPs at a regional level, although there are inconsistencies in strain relationships, for instance in the position of ‘TvGordo’ and ‘TvMagna’, which may reflect sampling error within the SNP tree or ancestral gene flow between *T. vivax* populations. Overall, VAP broadly reflects both population history and location.” (p5, ll116-122).

9. *Line 146 a COG is defined as a cluster of orthologs based on $\geq 90\%$ sequence identity, so “if they were relatively recent gene duplications” doesn’t make sense, unless each COG comprises 2 genes.*

This can make sense since a COG might consist of orthologous VSG with $\geq 90\%$ sequence identify, but from Nigerian strains only, in which case, one explanation (if the lineage is young relative to

cosmopolitan VSG with widespread distributions) is that this is a 'recent duplication' and derivation of a cosmopolitan lineage in Nigerian *T. vivax* only. This would offer a way in which populations could increase diversity without recombination. We tested whether the Nigeria-specific COGs were as old as their closest cosmopolitan COGs in the same phylotype or younger. They were the same age, leading us to the conclusion that the gene had been lost long ago from other populations, rather than gained in Nigeria. However, we have re-written the paragraph to make it clearer (p7, ll159-165).

10. Line 178-180 needs to be rewritten for clarity, defining the meaning of multi-, un- and fully coupled.

We have included a description of this nomenclature (p8, ll195-200) and also in the methods ((p26, ll643-650).

11. Line 240-241 "P24 is a dominant variant antigen" suggests a single entity, but next "the actual P24 transcript expressed was different in each case....diverged up to 26.5%", so there were multiple different P24 genes being expressed. Is there any evidence whether or not the antigens encoded are distinguishable immunologically? This is part of a larger problem for the non-specialist reader, as several different terms are used for grouping VSG genes - VAP, COG, phylotype, Fam – perhaps a Venn diagram to clarify this for the reader?

We have no evidence that the co-expressed VSG were immunologically distinct, this is the subject of our next project. Regarding the P24 comment, we have re-written the sentence to address this (p11, ll267-270). We have included the suggested Venn diagram in a new supplementary figure 2.

12. Line 247-8 delete one "only".

This has been corrected (p11, l276).

13. Line 256-7 "more centrally placed" - means mid infection?

Here, "centrally placed" related to the position in the phylotype network. We have changed the text to clarify this description (p12, l285).

14. Line 271-2 "phylogeny (sequence identity)" – these are not equivalent terms. Do you mean sequence relatedness? "VSG expression profile" = pattern of VSG expression over time?

We have replaced sequence identity for 'sequence relatedness' and VSG expression profile for 'pattern of VSG expression over time' (p12, ll299-300).

15. Lines 289, 293, 309, 314 "percentage of read-pairs that mapped to unpaired genomic positions", "the percentage of VSG read-pairs split after mapping", "these only implicated very closely related sequences", "therefore, assortment of *T. brucei* order was sort seen" – sentences of uncertain meaning that need rewriting for clarity.

We have re-written these sentences as suggested (p13, ll318-324).

16. Line 318 "The current model of trypanosome antigenic variation has recombination as the driver behind novelty and persistence" – clarify for the reader, e.g. novelty = the creation of novel VSG genes? Persistence = persistence of infection in the mammalian host?

Yes, we have changed the text to comply (p14, l346-7).

17. Line 346-7 *The reader needs more information here - what is the mechanism of VSG switching in T. brucei? What is known about expression sites in T. vivax?*

Agreed. We have included an explanation of the main mechanisms of VSG switching in *T. brucei* and the fact that telomeric expression sites have never been described for *T. vivax* (p16, l1400-7).

18. Line 362-3 *What T. vivax strains were used here? The assumption is that these workers used genetically similar strains of T. vivax, but this is not necessarily so.*

References 34 and 35 used different strains, from different geographical locations, but nevertheless still observed that *T. vivax* infections were became undetectable faster than *T. congolense*. We believe that this strengthens our assertion that propensity for self-cure is a species difference. The studies on acute syndromes (ref. 36 and 37) used the same *T. vivax* haemorrhagic strain (IL2337).

19. Line 380 *“Antigenic diversity in T. vivax is finite, in a way that T. brucei and T. congolense are not” - doesn’t make sense grammatically.*

We have rephrased it: Antigenic diversity is finite in *T. vivax*, but not in *T. brucei* and *T. congolense* ((p18, l1447-8).

20. Line 382 *“the lack of adaptation for persistence, so evident in T. brucei” – this seems to state the opposite of what was intended.*

We have changed this (p18, l451).

21. Line 446 *need a reference for Fam23-26.*

We have included this (p21, l519).

22. Line 496 *“infected with the T. vivax Lins isolate.” Is T. vivax Lins a clone? If not, this significantly alters interpretation of the animal infection data.*

All animals were infected with the same blood stabilate, collected from the same animal, at the same timepoint. Technically this is not a clone, as we would expect minor variants to be sampled when collecting the blood at peak parasitaemia. However, in the absence of a culture system for *T. vivax* bloodstream forms, this is the only possible way. We have made this clear in the results and methods:

“The *T. vivax* (Lins) inoculum was not derived from a clone, but rather represents a mixed population with one dominant clone (see Methods), and variation in VSG expression between animals could reflect this initial heterogeneity.” (p10, l252)

“There is no in vitro culture system for bloodstream-stage *T. vivax*. Therefore, it is not currently possible to derive single clones from blood, and the frozen stabilate used here represent mixed populations of the antigenic types circulating the donor animal prior to the experiment. However, we can expect one or two clones to be dominant within these populations and all animals received aliquots of the same preparation from one donor.” (p23, l576-580)

23. Line 499 Since Fig 4 graphs indicate that parasitaemia drops to zero, it is of interest how parasitaemia was determined. Details should be included, as the reference given is obscure (thesis?).

We have added a short description of the parasitaemia measurement method (p44, l1110-1). We have also replaced zero for “DL” in the parasitaemia graphs and noted the detection limit in the figure legend (4.1×10^3 trypanosomes/ml blood).

24. Line 508 It would be useful to mention the estimated number of trypanosomes from which RNA was derived.

RNA was derived from 2.23×10^5 to 1.04×10^7 trypanosomes, depending on the sample. We have included this information (p24, l597).

25. Fig 1. Legend - What is meant by “population history”? Figure – it is not obvious what the grey background shading on the left indicates. The colours for text are not readily distinguishable. Could use a 2 letter country code in the strain name instead?

Population history refers to genetic proximity. We have modified Fig. 1 and its legend to incorporate this feedback. Specifically, we have changed the labelling and colour scheme in the figure and re-written the legend (p42, l1055-69).

26. Fig 2. Legend – “global *T. vivax* repertoire” is misleading. Some of the colours are not readily distinguishable, e.g. for cosmopolitan and Nigeria.

We have removed the word global and changed the colour scheme of Fig. 2.

27. Fig 4. Line 829 “classical expectation” – surely the idea that each parasitaemic peak represents a single VSG was realised to be simplistic several decades ago? Indicate on the figure that the graphs pertain to animals 1-4. The parasitaemia is shown decreasing to zero, but there is likely a cut-off of parasite detection above zero. The graphs of detected phylotype are shown as curves, but the RNA data derive only from the peaks of parasitaemia, so what’s the justification for showing the data as curves?

We have added the animal number in the figure and have replaced zero by the detection limit (already addressed in comment 24).

Whilst it is true that the “classical expectation” is no longer that each parasitaemia peak represents a single VSG, it is still thought to include a single superabundant (i.e. predominant) VSG, which is one of the reasons why there is a fast and partially effective antibody-driven parasite clearance. In this particular sentence, we were referring to the concept of superabundant VSG, rather than one VSG per peak. In light of this comment, we have rephrased the text to clarify this (p44, l1125).

Regarding the phylotype graphs, as with any line graph that samples a continuous process discontinuously, we understand that by drawing curves we are interpolating data that we did not see, but in the absence of constant, real-time expression data, we have no evidence to suggest any other pattern. This visualization was used in previous studies describing antigen expression patterns (e.g. Mugnier *et al.* 2015) and we believe it remains the easiest way to depict phylotype expression rise and fall over time.

Yours sincerely,

Dr Andrew Jackson Dr Sara Silva Pereira

Reviewers' Comments:

Reviewer #1:

Remarks to the Author:

I read in detail the rebuttal and the new version of the paper by Pereira and colleagues. The authors have responded adequately to all concerns raised by me and the other reviewers. As it stands, I don't see major issues that would prevent the publication of this work.

While reading the manuscript, I found these minor edits:

Line 134: should this be Supplementary Figure 2 instead of Supplementary Figure 1?

Lines 184-186. The definitions of FC, MC and UC given in the results section does not entirely match the definitions given in the methods. In the results section, it is said that donors are classified based on number of fragments but also coverage of the VSG. In the methods section, it is only based on number of donors. In addition, in the results section it is written that "MC VSGs are sequences with donor(s) contributing to less than >84% of the sequence but more than 150bp". I guess you mean "less than 84%", not "less than >84%"? In addition, with "more than 150bp" you probably mean at least 2 donor fragments; just to be consistent with the definition for UC. If indeed at least two 2 fragments, then the example of figure 3b for MC is not entirely correct as donor 2 has only 1 fragment. This is not a major problem, but I think the definitions should align throughout the text and figures.

The order of the figures at the end of the paper seem to be switched.

Dr. Frederik Van den Broeck

Reviewer #2:

Remarks to the Author:

I am happy with the reviewers responses to my comments and have no further issues with this manuscript.

Reviewer #3:

Remarks to the Author:

The authors have made a thorough revision of the MS addressing all reviewers' points comprehensively. Much improved MS.

Still a few grammatical errors, mostly non-matching singular/plural noun and verb, which will get picked up by CE anyway, e.g. line 409-10,544-5,902,905. Confusion of transcript and protein needs to be corrected "all VSG transcripts are serologically distinct" line 364. Qualify "genetic repertoire" line 48.

Fig legends now direct to "source data files" - will these be identified?

Reviewer #1:

1. *Line 134: should this be Supplementary Figure 2 instead of Supplementary Figure 1?*

Yes, we have changed this. Thank you.

2. *Lines 184-186. The definitions of FC, MC and UC given in the results section does not entirely match the definitions given in the methods. In the results section, it is said that donors are classified based on number of fragments but also coverage of the VSG. In the methods section, it is only based on number of donors. In addition, in the results section it is written that "MC VSGs are sequences with donor(s) contributing to less than >84% of the sequence but more than 150bp". I guess you mean "less than 84%", not "less than >84%"? In addition, with "more than 150bp" you probably mean at least 2 donor fragments; just to be consistent with the definition for UC. If indeed at least two 2 fragments, than the example of figure 3b for MC is not entirely correct as donor 2 has only 1 fragment. This is not a major problem, but I think the definitions should align throughout the text and figures.*

If a VSG has two donors, each contributing to 150bp only, but at different regions (i.e., not overlapping), we consider them MC, not UC, because they we have evidence for multiple donors. We have edited the text in lines 184-186 to make this clear: "MC VSGs are sequences with donor(s) contributing to less than 84% of the sequence but more than 150bp, or at least 2 donor fragments in different regions."

3. *The order of the figures at the end of the paper seem to be switched*

Figures 3 and 4 were switched in the PDF version of the manuscript, but the naming is correct.

Reviewer #3:

1. *Still a few grammatical errors, mostly non-matching singular/plural noun and verb, which will get picked up by CE anyway, e.g. line 409-10,544-5,902,905.*

Apologies if this is so, but we cannot identify the errors from the comment. We will be happy to correct them should they be described.

2. *Confusion of transcript and protein needs to be corrected "all VSG transcripts are serologically distinct" line 364.*

We have corrected this, it now reads "if all VSG transcripts are represent serologically distinct proteins"

3. *Qualify "genetic repertoire" line 48.*

We have added the qualification "particularly with regard to cell surface-expressed genes" to reflect what is contained in the study cited.

4. *Fig legends now direct to "source data files" - will these be identified?*

Yes. Each sheet in the Excel source data file has the identification of which figure it relates to.

Yours sincerely,

Dr Andrew Jackson Dr Sara Silva Pereira